# Use of Antibiotics in Preterm Newborns

**DOI:** 10.3390/antibiotics11091142

**Published:** 2022-08-23

**Authors:** Raffaele Simeoli, Sara Cairoli, Nunzia Decembrino, Francesca Campi, Carlo Dionisi Vici, Alberto Corona, Bianca Maria Goffredo

**Affiliations:** 1Division of Metabolic Diseases and Drug Biology, Bambino Gesù Children’s Hospital, IRCCS, 00146 Rome, Italy; 2Neonatal Intensive Care Unit, University Hospital “Policlinico-San Marco” Catania, Integrated Department for Maternal and Child’s Health Protection, 95100 Catania, Italy; 3Neonatal Intensive Care Unit, Medical and Surgical Department of Fetus-Newborn-Infant, Bambino Gesù Children’s Hospital, IRCCS, 00165 Rome, Italy; 4ICU and Accident & Emergency Department, ASST Valcamonica, 25043 Breno, Italy

**Keywords:** pharmacokinetic, pharmacodynamic, antibiotics, preterms, therapeutic drug monitoring, micro-sampling

## Abstract

Due to complex maturational and physiological changes that characterize neonates and affect their response to pharmacological treatments, neonatal pharmacology is different from children and adults and deserves particular attention. Although preterms are usually considered part of the neonatal population, they have physiological and pharmacological hallmarks different from full-terms and, therefore, need specific considerations. Antibiotics are widely used among preterms. In fact, during their stay in neonatal intensive care units (NICUs), invasive procedures, including central catheters for parental nutrition and ventilators for respiratory support, are often sources of microbes and require antimicrobial treatments. Unfortunately, the majority of drugs administered to neonates are off-label due to the lack of clinical studies conducted on this special population. In fact, physiological and ethical concerns represent a huge limit in performing pharmacokinetic (PK) studies on these subjects, since they limit the number and volume of blood sampling. Therapeutic drug monitoring (TDM) is a useful tool that allows dose adjustments aiming to fit plasma concentrations within the therapeutic range and to reach specific drug target attainment. In this review of the last ten years’ literature, we performed Pubmed research aiming to summarize the PK aspects for the most used antibiotics in preterms.

## 1. Introduction

Generally, neonatal pharmacology tends to treat preterm and full-term newborns as the same subjects. However, preterms are physiologically and pharmacologically different from full-term neonates [1,2]. Usually, these differences are, at least in part, reduced by using the post-menstrual age (PMA) instead of the post-natal age (PNA), since it considers gestational and maturational aspects involving the different pharmacokinetic (PK) phases of ADME (Absorption, Distribution, Metabolism, and Excretion) [3]. However, not all the differences between term and preterm neonates can be normalized using this linear proportion. In fact, some specific processes, such as gastric emptying, renal, hepatic, and intestinal functions, are significantly different in preterms compared to full-term neonates [4,5]. Therefore, PK studies should be mainly focused on this special population; however, sampling limitations, subject vulnerability, and ethical concerns do not allow researchers to carry out studies on pediatric patients and neonates. The result of this paucity of studies on newborns is that almost 40–80% of drugs used in Neonatal Intensive Care Units (NICUs) are off-label or unlicensed for this population [6]. This means that dosing strategies on newborns are translated from clinical trial data performed on adults [7]. Another limiting aspect is that the collection of large number of samples to perform individual PK models is not feasible in children and especially in neonates [8]. To overcome this issue, allometric scaling has been proposed to translate pediatric dosing from adult studies, but a single allometric scaling model that can fit all drugs and all age groups is not always available [9]. Therefore, population pharmacokinetic (popPK) and physiologically based pharmacokinetic (PBPK) models seem to be the ideal strategy for the neonatal population, for whom the repeated sampling of high volumes of blood is not physiologically and ethically acceptable [8,10,11]. Moreover, this approach, differently from allometric scaling, which only takes the size of the subjects into account, allows researchers to consider multiple patients’ related covariates [12]. A recent review from van den Anker J. and Allegaert K. (2021) focused on how to translate the results of popPK studies into a rational dosing regimen and how PBPK modeling, electronic health records, and the abundantly available data of vital functions of premature newborns can be used to optimize the most safe and effective dose in these neonates [13].

In neonatal populations, anti-microbial agents have been widely studied by using these modeling approaches [12]. In fact, perinatal and hospital-acquired infections are a serious issue among neonates, especially those admitted to NICUs [14]. Moreover, infections that occur at this life stage can severely compromise not only survival but also the neuro-developmental outcomes of newborns [15]. Therefore, the prevention and treatment of microbial infections are of pivotal importance in this special neonatal setting. Preterms are particularly susceptible to microbial infections since they are often subjected to invasive procedures, including the need for central catheters for parental nutrition and ventilators for respiratory support [15]. Sometimes, especially during long hospitalization periods, these life-saving devices can become a source of microbes and an appropriate shunt-lock therapy is, therefore, required to avoid systemic infections [16]. As previously stated, one of the main limits of performing clinical studies on neonates is represented by the physiological and ethical concerns related to sampling procedures. For this purpose, several micro-sampling techniques have been used to draw blood samples from neonates. Since their introduction in 1960, dried blood spots (DBS) have been one of the most used sampling strategies for blood collection in neonates to screen babies for phenylketonuria (PKU) and to monitor dietary treatment [17]. DBS are widely used within newborn screening (NBS) programs for the diagnosis of inherited metabolic disorders (IMDs). However, in the last few decades, a growing interest has been registered about the use of DBS for therapeutic drug monitoring (TDM), which consists of measuring the concentrations of drugs at specific time points to prevent adverse reactions (especially for drugs with a narrow therapeutic window) or to individualize therapy according to patients’ needs [18]. It is worth noting that, in this context, an accurate quantification of drug concentration is desirable, since a deviation from true values may significantly affect clinical decisions. Therefore, the use of DBS for performing TDM has been replaced by alternative micro-sampling techniques, which are less sensitive to the well-known hematocrit (Hct) effect [19,20]. Among these techniques, volumetric absorptive micro-sampling (VAMS) devices were introduced in 2014 as Mitra^®^ (Neoteryx LLC, Torrance, CA, USA). They consist of a plastic rod ending with a hydrophilic absorbent tip. One of the main advantages of these tools is the possibility to collect fixed volumes of blood (10, 20, or 30 µL) from a finger-prick by capillarity and without any need for cannulas or syringes [21,22]. Another important aspect is that VAMS allow researchers to overcome the hematocrit effect that is often associated to DBS sampling and that may affect the quantitative analyses of several drugs [21,22]. So far, Mitra^®^ has already been used as sampling strategy to conduct different pharmacokinetic studies [23,24,25,26].

In this review of the last ten years’ literature, we will specifically focus on the pharmacokinetic aspects of antibiotics used as both prophylactic and therapeutic treatments of microbial infections in preterm newborns. In particular, we will classify the most used antibiotics, reporting the different posologies and the administration routes adopted. Moreover, we will describe the most recent findings on popPK modeling studies aimed at evaluating antibiotic administration in preterms, as well as the latest dosing strategies suggested. Finally, we will discuss the use of micro-sampling strategies during the pre-analytical phase and the main bioanalytical techniques applied to the TDM of antibiotics in preterm neonates.

## 2. Definition of Preterm Neonates

The World Health Organization (WHO) defines preterms as babies born alive before 37 weeks of pregnancy are completed. There are sub-categories of preterm birth based on gestational age (GA): extremely preterm (less than 28 weeks), very preterm (28 to 32 weeks), and moderate to late preterm (32 to 37 weeks).

It has been estimated that almost 15 million babies are born too early every year. That is more than 1 in 10 babies. Approximately 1 million children die each year due to complications of preterm birth [27], meanwhile many survivors face a lifetime of disability, including learning disabilities and visual and hearing problems. Globally, prematurity is the leading cause of death in children under the age of 5 years. Additionally, in almost all countries with reliable data, preterm birth rates are increasing. The survival rate significantly changes across countries. In fact, in poor countries, half of the babies born at or below 32 weeks (2 months early) die due to the lack of feasible, cost-effective care, such as warmth, breastfeeding support, and basic care for infections and breathing difficulties. Conversely, in industrialized countries, the majority of these babies survive [28]. 

Preterm birth occurs for a variety of reasons. Most preterm births happen spontaneously, but some are due to early induction of labor or cesarean birth for mother-dependent reasons. Common causes of preterm birth include multiple pregnancies, infections, and chronic conditions such as diabetes and high blood pressure (both chronic and induced by pregnancy); however, often no cause is identified. There could also be a genetic influence. An improved understanding of the causes and mechanisms will advance the development of solutions to prevent preterm births.

The WHO has developed new guidelines with recommendations for improving the outcomes of preterm births. This set of key interventions can improve the chances of survival and the health outcomes for preterm neonates. The guidelines include interventions provided to the mother—for example steroid injections before birth, antibiotics for premature ruptures of membranes, and magnesium sulphate to prevent any future neurological impairment of the child—as well as interventions for the newborn baby—for example thermal care, early feeding support, kangaroo mother care, safe oxygen use, surfactant administration, and strategies to reduce invasive mechanical ventilation implementing not invasive respiratory support (WHO recommendations on interventions to improve preterm birth outcomes).

## 3. Physiological Aspects of Preterm Neonates

Normally, full-term pregnancy is characterized by specific developmental processes that, before delivery, allow the fetus to reach a maturation state sufficient to guarantee its transition from uterine to extra-uterine life. In premature neonates, these processes are not fully completed, therefore preterm newborns can experience difficulties after birth [29]. Physiologically, this transition depends on several factors that include maternal medical history, duration of pregnancy, placental condition, and presence of fetal congenital anomalies [30]. As consequence, the physiological characteristics of preterms can significantly affect the pharmacokinetic (PK) and pharmacodynamic (PD) behavior of different drugs, making these subjects a special population that deserves specific pharmacological consideration [31].

Thermal regulation is a pivotal process for human beings and, in neonates, is the result of several factors, including gestational and postnatal age, body weight (BW) at birth, and neonate clinical conditions. In newborns, the head represents almost the 20% of total surface area and is the main source of heat loss. Preterms are characterized by thinner skin compared to full-term neonates, and, similarly, the subcutaneous fat layer is not adequately developed to prevent evaporative heat loss [31]. Since shivering is a physiological process gradually acquired after the first year of life [32], newborns adopt a non-shivering thermogenesis based on brown-fat deposition and oxygen consumption.

Glucose regulation is another critical process during the transition to extrauterine life. After birth, the balance between glucagon and insulin shifts toward the former; therefore, glycogenolysis, gluconeogenesis, and lipolysis are activated in order to provide enough energy to the neonate [29]. Almost 15% of full-term neonates present hypoglycemia, however, preterms show a higher risk of developmental delays as a consequence of impaired glycogen, proteins, and fat storage, which usually happens during the third trimester of pregnancy [29]. Therefore, lipolysis represents the main source of energy when glycogen storage is inadequate. In fact, glycerol or ketone bodies are an important fuel substrate in these particular conditions. However, preterms are characterized by a limited fat deposition that does not allow a high rate of lipolysis or ketogenesis processes [33]. On the other hand, hyperglycemia is mainly recurrent in neonates with very low birth weights, especially during the first week of life, and in newborns who also present growth restrictions or receive glucose within parenteral nutrition. It was previously reported that almost 80% of very-low-birth-weight neonates present with hyperglycemia [33,34]. This hyperglycemia is the result of several processes that mainly include an absence of gluconeogenesis inhibition and the presence of insulin resistance (perhaps as a consequence of high growth hormone, cortisol and catecholamine release, or defective proinsulin processing and secretion). Uncontrolled hyperglycemia can lead to serious consequences, such as hyperosmolality, dehydration following to osmotic diuresis, and intracranial bleeding [33,34].

Respiratory function is an important process that undergoes significant changes during intrauterine development. In particular, pulmonary surfactant, which reduces the surface tension in the alveoli facilitating the respiratory mechanism, is produced during the 34th week of gestation. Often, preterms are characterized by an impaired or altered surfactant production that leads to respiratory distress syndrome (RDS) [35]. Apnea of prematurity, which is defined as the absence of breathing for 15–20 s, is another common feature in almost 85% of neonates born before 34 weeks and is believed to be a consequence of immaturity of the central respiratory control systems [36]. 

The ability of the immune system to respond to pathogenic insults develops during the first three months after delivery [37]. In fact, the immune system development during the uterine life takes place in a sterile environment without any antigenic exposure. In particular, the complementary system, an important component of both the innate and adaptive immune system, is already underdeveloped in healthy full-term neonates [38]. Therefore, preterms are more susceptible to the intracellular pathogens that are responsible for both intra-partum and post-partum infections [37]. Moreover, preterms show low levels of regulatory T cells (Treg), leading to an over-reactivity of their immune systems [39]. 

Another physiological aspect that often characterizes preterm neonates is the low production of the adrenocorticotropic hormone (ACTH). In fact, a fetus does not produce cortisol until 30 weeks of gestation [29]. Therefore, preterms show an altered response to stress factors, although this adrenal gland deficiency is limited to the first two weeks of life and is therefore defined as a temporary adrenocortical insufficiency.

Finally, renal and hepatic functionalities are strictly related to the maturation degree of both the kidneys and liver. In particular, the number of mature nephrons is reached around the 36th week of gestation. Therefore, preterms could be affected by a reduced or altered nephrogenesis process compared to full-term neonates [29]. In fact, preterms show a significantly lower renal flow rate compared to full-terms, and this aspect leads to a limited ability to perform glomerular filtration and an impaired capability to avoid liquid accumulation [40]. Similarly, liver function follows several maturational steps during intrauterine life and even after birth; although functionality is not fully reached in newborns, they rarely experience serious long-term complications. Conversely, preterms are more susceptible to developing serious consequences of an immature liver. These include hypoglycemia, hyperbilirubinemia, cholestasis, bleeding, and a lower ability to metabolize drugs [41].

## 4. Pharmacological Considerations in Preterm Newborns

The efficacy and safety of pharmacotherapy in neonates must consider the PK/PD properties of a specific drug together with the physiological characteristics of these patients [42,43]. Similarly, the physiological aspects that characterize preterm newborns are able to directly interfere with both the PK and PD of many administered drugs. Pharmacokinetics are usually described as “what the body does to a drug”. More specifically, it is a branch of pharmacology aimed at studying the fate of pharmacological substances in the body throughout several phases that include drug absorption, distribution, metabolism and elimination (ADME).

Oral drug absorption trough the gastrointestinal tract is usually impaired in preterms due to several factors that include a reduction in gastric and bile acid secretion and a lowered gastrointestinal motility [2,44]. In particular, gastric acidity is able to affect the ionization degree of many molecules, especially the weak acids, which, at pH values higher than the dissociation constant (pKa), are present in their dissociated structure. This aspect reduces the amount of the drug able to cross the membranes and, indeed, the oral bioavailability. Similarly, the lower production of bile and pancreatic enzymes typical of preterms reduces the extent of the absorption of fat-soluble drugs. Gastrointestinal motility and the maturation of intestinal mucosa are important factors influencing drug absorption. In particular, intestinal villi reach full maturation around 20 weeks of gestation [45]. As a consequence, preterms are endowed with a reduced surface area due to a limited villi maturation and intestinal mucosa development, which results in an impaired ability to absorb drugs via the gastrointestinal tract compared to full-term neonates. Conversely, drug administration through the percutaneous route requires, in preterms, a dose reduction due to the high weight-to-body surface area ratio. In fact, although the muscle mass is generally lower in neonates compared to adults, the degree of muscular vascularization is much higher in these subjects as a compensation mechanism. Therefore, topically administered drugs could be characterized by a higher degree of absorption through the percutaneous route compared to adult patients [46].

The body’s water composition changes alongside aging, and the 80% total body water in preterms is reduced to 70% in full-term neonates, reaching almost the 60% in adults [47]. Therefore, the extracellular fluid composition is higher in neonates than in adults. Similarly, the percentage of fat deposition tends to be smaller in preterms (almost 3%) compared to the 12% seen in full-terms [48]. As consequence, the distribution volume (Vd) for many drugs could be different in preterm neonates and could change with aging. For example, hydrophilic drugs could be much more heavily distributed in preterms rather than fat-soluble molecules due to the different composition of the body’s water and fat in these subjects [49]. A similar concept can be applied to the different composition of plasma proteins in newborns compared to older infants and adults [50]. In fact, the total protein, albumin, and alpha-1-acid glycoprotein levels are lower in preterms, affecting the capacity of many drugs to bind proteins [49]. Therefore, in conditions of hypoalbuminemia, drugs with a high percentage of protein binding could show an increased plasma concentration of the “free” (protein unbound) fraction, leading to a higher risk of toxicity.

The ability to metabolize drugs is strongly affected by age [51]. The liver is the main route of drug metabolism, and an alteration or impairment of this capability can lead to drug accumulation in the body with a raised chance of adverse events. Conversely, pro-drugs that need a metabolic process in order to release the active molecule could show a reduction in their pharmacological effects, since the circulating levels of active compound are significantly low. Drug metabolism is usually divided into two different phases: phase I and II. The players who act in these two steps undergo a maturation process that is age dependent [52]. For example, cytochrome (CYP) P450 enzymes are involved in the phase I reaction, and their maturation usually starts around the 25th week of gestation [53]. In fact, it has been demonstrated that CYP enzymes’ quantity in fetal livers is around 30–60% of adult values, and that the full CYP ability is usually reached by 2 years of age [46,53]. An impaired activity of these immature enzymes leads to serious toxicities, usually registered in premature neonates. Typical examples are grey baby syndrome following chloramphenicol administration or gasping syndrome after benzyl alcohol accumulation [54,55,56]. In the grey baby syndrome, phase II uridine diphosphate glucuronosyltransferase (UDP) isoenzymes are involved; their maturation follows different steps and tends to be downregulated until postnatal day 10 [57]. It is worth noting that liver maturation of phase II enzymes is less known compared to cytochromes [46]. However, the maturation of UDP-glucuronosyltransferase (UGT) has both PK and PD implications, such as in chloramphenicol toxicity and morphine glucuronidation, respectively [57]. Therefore, the efficiency of drug metabolism, as well as the chance of drug–drug interactions (DDI), should be considered in preterms, especially when hospitalization into intensive care units (ICUs) requires multiple concomitant therapies.

The final step of the pharmacokinetic process is represented by excretion. It consists of the removal of a drug and/or its metabolites from the body and usually happens through the renal or hepatic route, although other alternative ways are available, depending on the drug’s chemical–physical characteristics [52]. As previously stated, nephrogenesis is complete around the 34th week of gestation, even if several factors can affect the maturational process. For example, drugs such as anti-inflammatory consumed during pregnancy can be nephrotoxic and can induces delays in kidney development [58,59,60]. Even during the PNA, some drugs, such as amphotericin-B and the aminoglycoside antibiotics, can induce renal damages; therefore, their utilization requires the tight monitoring of renal functionality. In this regard, the glomerular filtration rate (GFR) reaches half of the adult value by 3 months of age and the adult levels by 2 years of age [61]. Preterms or neonates who are born small for their gestational age are characterized by impaired renal excretion, and the tubular secretion is not adequate because of a reduced perfusion rate [62,63]. In fact, preterm neonates show a maturation of tubular function that is slower during the first weeks of life compared to the full-terms, and the maximal GFR is achieved at week 48 of normal GA (2–4 mL/min in full-terms and 0.6–0.8 mL/min in preterms) [45,64]. Finally, it is worth noting that creatinine at birth is not the ideal marker for glomerular filtration, because, in the first few hours of life, it reflects the mother’s renal function [65]. However, it becomes more reliable after the first weeks of post-natal life [66].

Alongside the maturational factors, there are also disease-related PK alterations that need to be considered. These factors include critical illness, inflammatory status, and augmented renal clearance (ARC). In this context, a recent review of the literature illustrates the impact of both maturational and non-maturational factors on absorption, distribution, metabolism, and excretion applied to antibiotics [14]. Table 1 summarizes the main physiological aspects that differentiate preterms from full-term neonates alongside their effects on PK processes.

Pharmacodynamics (PD) are often defined as “what a drug does to the body”. It is the branch of pharmacology that studies the biochemical and physiologic effects of drug administration. More generally, PD studies also involve the evaluation of the drug’s receptor binding (including receptor sensitivity), post-receptor effects, and chemical interactions. These considerations help to define the dose–response relationship and, therefore, the drug’s effects. In fact, the pharmacological response depends on the ability of each drug to reach the site of action at a concentration sufficient to bind its target and to induce receptor activation. Therefore, developmental changes associated with fetus maturation during the uterine life and then with newborns in the post-natal phase involve not only the drug’s PK, but also the study of their mechanism/s of action. In fact, target receptors are proteins that follow age-dependent maturation processes, and, indeed, it is not surprising to find that the pharmacological effects of many drugs are conditioned in preterm newborns, not only due to pharmacokinetic aspects, but also due to an incomplete maturation of PD mediators.

In this regard, a recent review from Machado JS and colleagues (2021) reports several examples of specific drug classes whose actions are affected by the physiological changes observed in preterm neonates. For example, prokinetic agents may not be completely effective in preterms and only partially effective in full-terms due to an age-dependent expression of intestinal motilin receptors that regulate antral contractions [31]. Similarly, bronchodilator agents are ineffective in preterms because of the paucity of bronchial smooth muscles. These concepts can also be extended to the adverse events that could be induced by the administration of specific drug classes in preterm neonates. For example, calcium channel blockers (CCB) can lead to life-threatening bradycardia and hypotension since cardiac calcium deposits in the endoplasmic reticulum are reduced in the neonatal heart, and the exogenous calcium has a prevalence toward cardiac contractility in these subjects [31].

**Table 1 antibiotics-11-01142-t001:** Physiological aspects of preterm neonates affecting PK processes.

PK Process	Physiological Hallmark	Pharmacological Implication	Reference
Absorption	Reduction in gastric acid secretion	Reduced oral bioavailability of weak acids	[2,44]
Absorption	Reduction in bile acid secretion	Reduced oral bioavailability of fat-soluble drugs	[2,44]
Absorption	Impaired gastrointestinal motility and low maturation degree of intestinal mucosa	Impaired ability to absorb drugs via gastrointestinal tract	[45]
Absorption	High extent of cutaneous vascularization	Higher degree of absorption trough the percutaneous route	[46]
Distribution	High body water composition	Hydrophilic drugs could be much more distributed	[49]
Distribution	Low percentage of fat deposition	Limited distribution of fat-soluble molecules	[49]
Distribution	Hypoalbuminemia	Increased plasma concentration of “free” (protein unbound) drugs’ fraction	[49]
Metabolism	Impaired maturation of phase I and II enzymes	Reduced drugs’ metabolism and/or pro-drug activation	[57]
Excretion	Uncompleted nephrogenesis and reduced renal perfusion rate	Impaired renal excretion and lower tubular secretion for drugs eliminated through kidneys	[62,63]

## 5. Population Pharmacokinetic (popPK) Modeling Approaches in Preterm Infants

Researchers who are planning to carry out pharmacokinetic studies on neonates should be aware of the challenges that must be faced. These barriers include ethical concerns and approval, an appropriate study design, and the chance of measuring drug concentrations in multiple blood samples [65]. In fact, the development of traditional PK models, aimed at obtaining dose information, requires, for each recruited patient, several consecutive blood drawings in order to establish a temporal profile of plasma drug concentrations. This approach is not always feasible for children and neonates due to the ethical considerations that limit the number of samples and the total blood volumes collected. To overcome this issue, several routes can be followed. One of these consists of using leftover blood samples previously used for other clinical analyses. This approach has already been adopted and avoids additional blood sampling [67,68].

Another strategy is represented by the use of pharmacometric models and simulations that allow researchers to evaluate PK behavior and personalize drug dosing in neonates [11,52]. Pharmacometrics consist of developing and applying mathematical and statistical methods in order to characterize and predict a drug’s PK and its effect on clinical responses [69]. Population pharmacokinetic models (popPK) represent a valid example of the pharmacometric approach. They have already been used to study the PK behavior of different drug classes, including antibiotics, as reviewed by Wilbaux M and colleagues [70].

These models use concentration values obtained from several clinical subjects to predict the PK behavior of drugs in that population. Moreover, through this approach, a minimum number of samples can be established for each subject [71]. Additionally, popPK models allow researchers to include disease-specific changes in pharmacokinetic parameters that could affect drug dosing. These data include not only patients’ characteristics, such as age (both post-natal and post-menstrual), weight, and height, but also information on renal and hepatic functions. These studies can be performed using different approaches that include the naïve pooled data approach, the standard two-stage approach, and the mixed effect models [72]. Mixed-effects population pharmacokinetic models are particularly useful for studying neonatal PK, since they consider the population as the unit of the analysis, rather than the individual [65]. In fact, mixed-effects models allow researchers to use a small number of samples from a large number of subjects. This aspect limits the number of samples required from each subject and is particularly relevant when working with neonates due to the ethical considerations involved. Another important difference of traditional PK studies is that the adherence to specific sampling time points is not mandatory, and, therefore, leftover samples from routine clinical analyses can be used under real-life conditions, and this is an additional benefit since it limits additional blood sampling [7,12].

The mixed-effects modeling approach has already been applied to neonatal data [71,73,74]. In this approach, a nonlinear mixed-effect modeling is used to fit a compartmental model to in vivo PK data. To explain inter-individual variability in PK data, the structured compartmental model is expanded with a stochastic and a covariate model. In fact, a mixed-effects modeling approach allows the data to be explained using a mixture of fixed and random effects. The stochastic model describes the random variability among individuals for the PK parameters that cannot be predicted from an average of fixed covariates but are usually assumed to be a lognormal distribution of these parameters within the population, whereas fixed effects are applied to explain the inter-individual variability in a specific PK parameter using demographic and other characteristics (covariates) of the studied subjects, such as age, sex, weight, and creatinine clearance. Therefore, these covariates can be accounted for as fixed effects, since they are considered to be predictive of the inter-individual variability [75,76,77]. Although it is not specifically mandatory, the determination of the optimal time for sampling could be helpful to reduce the number of blood drawings. In this context, optimal designs (D-optimal) are a form of design provided by a computer algorithm and can be used within neonatal studies to determine the best sampling time to be incorporated into a model. Therefore, this approach allows researchers to establish not only the number, but also the timing for the samples required to set up an ideal database for modeling creation [78,79]. It is worth noting that popPK models are still limited in preterm neonates due to the complexity and the limited knowledge of the developmental changes in drug-metabolism pathways, which take place after birth. Therefore, the ability of the model to predict the inter-individual variability as well as the pharmacodynamic effects is not fully estimated [80]. However, although popPK models are characterized by conceptual shadows and still represent an evolving field, they are a valid support for predicting the effects of drug administration on preterm neonates in the absence of traditional PK studies.

Allometric scaling has often been used to describe the PK in pediatric and neonatal populations by translating PK parameters from adults to children [81]. This approach assumes the existence of a linear relationship between body size and the physiological processes studied [82]. Within popPK modeling, the allometric relationships are based on the following equation:$$\theta_{i}=\theta_{ave}\times {\left(\frac{Size_{i}}{Size_{ave}}\right)}^{exp}\;$$
where *θi* is the PK parameter (i.e., Clearance or Vd) for an individual (*i*), and *Size_i_* is the body weight or the body surface area for that specific subject. Similarly, *θ**_ave_* and *Size_ave_* are the parameters for an average individual (for example an adult 70 kg male). The *exp* applied to the ratio depends on the PK parameter analyzed. For example, the scaling of Vd and blood volumes uses 1 as *exp*, meanwhile the clearance uses an *exp* = 0.75. The main limit of these allometric approaches for the scaling of PK parameters to neonates is the maturational processes that should be considered for these subjects. Especially for drugs’ clearance evaluations, the maturation of enzymes and transporters, as well as renal filtration mechanisms, are not completely mature in neonates. As a consequence, this aspect will have an influence on the calculation of clearance parameters by using allometric scaling. Therefore, to overcome this issue, maturation functions have been developed. These functions can be linear, exponential, or sigmoidal according to the range of ages they have to cover [83]. Combining these maturation functions with body-size-based scaling represents a successful approach, although, in several neonatal settings, more complex functions or specific databases are required to scale the PK parameters through the whole age range [84,85]. However, the complex maturational phases that involve ADME processes and that take place in preterm and full-term neonates, cannot be easily included in a single maturation function, making the allometric scaling’s reliability poor for these subjects.

### Physiologically Based Pharmacokinetic (PBPK) Modeling to Predict PK Parameters in Preterm Infants

Alongside popPK models, another approach that could be used to evaluate PK parameters in neonates is represented by the physiologically based pharmacokinetic (PBPK) modeling that allows an in vitro–in vivo extrapolation and follows the incorporation of several pathophysiological factors as system parameters [86]. This aspect is not easily applicable when using a compartmental popPK modeling approach [87,88]. PBPK models have recently gained attention as an ideal methodology to predict PK parameters in neonates [89,90]. This approach is based on the combination of drug-specific information (such as physicochemical characteristics of the drug) with physiological and anatomical knowledge into a predictive model. In fact, thanks to the combination of physiological parameters that involve ADME processes and the maturational degrees of organs and enzymes, the expected PK parameters can be calculated for different ranges of age using the same model [91,92]. Therefore, PBPK could help to predict the PK behavior of drugs in neonates and to evaluate the effects of both intrinsic (e.g., organ dysfunction, age, genetics) and extrinsic (e.g., drug–drug interactions) factors on drug exposure [90,93]. Moreover, PBPK also allows researchers to predict drug concentrations and their PK behavior in different tissues such as drug target sites [94]. The aim of PBPK is to use mathematical models (using ordinary differential equation) to describe the most important underpinning pathophysiological, physiochemical, and biochemical processes that affect the ADME behavior of the studied drug. In fact, a full PBPK distribution model adopts several time-based differential equations in order to simulate the concentrations in various organ compartments such as blood (plasma), adipose tissue, bone, brain, gut, heart, kidney, liver, lung, muscle, pancreas, skin, and spleen. The inter-individual variability is introduced through tissue-volume predictions, considering age, sex, weight, and height as covariates using a Monte Carlo sampling that takes into account the correlations between these covariates [95]. Therefore, PBPK could be a useful tool for studying drug PK in vulnerable subjects such as preterm neonates, for whom clinical studies are not always applicable. Moreover, the integration of popPK data with PBPK modeling should be preferred to obtain more robust results and to improve PK knowledge of this special population.

## 6. PK of Antibiotics in Preterms

Antibiotics are widely prescribed and administered in preterm neonates. Almost 61.3% of neonates admitted into neonatal intensive care units receive an antibiotic course during hospitalization. Nevertheless, most antibiotics were not investigated in neonatal PK studies before licensing and, therefore, are used off label. Both popPK and PBPK models have been developed so far to predict PK behavior and to tailor the dosing regimens of several antibiotics in neonates including preterms [96]. Recently, in order to achieve an appropriate antimicrobial stewardship, the urge for a precision medicine towards special patients’ populations, including late preterm and term neonates, is trying to overcome the “one-size-fits-all” approach based on general protocols and standard antibiotic treatment regimens [97]. A summary of PK studies based on antibiotic use in preterm neonates is reported in Table 2.

### 6.1. Aminoglycosides

Aminoglycosides are the antibiotics of first choice for the treatment of neonatal infections due to Gram-negative bacteria. Gestational age, postnatal age, birth weight, maturation of renal function, and the percentage of body water are factors that have a strong influence on the PK/PD behavior of these drugs [70]. Aminoglycosides exert their antibacterial action by interfering with bacterial protein synthesis. Since this class of antibiotics is characterized by renal elimination (until >90%), it must be considered that full nephrogenesis is completed in the third trimester of pregnancy [98]. Considering this aspect, previous studies performed on aminoglycoside antibiotics have shown that a dosing regimen based on a single higher daily dose and longer intervals should be preferred in order to guarantee the same therapeutic efficacy with reduced side effects [99,100]. Recently, Lee SY et al. (2021) conducted a study on 30 preterm neonates, demonstrating that acute kidney injury during aminoglycosides treatment is more severe at both lower gestational ages and birth weights [101]. The bactericidal activity of aminoglycosides against Gram-negative infections, together with their synergism with beta-lactam antibiotics, the limited bacterial resistance, and the convenient costs, has justified the wide use of these antibacterials in neonates [102]. The most used aminoglycosides in preterm infants are netilmicin, gentamicin, and amikacin.

Netilmicin is generally the first aminoglycoside used in neonates, both as a prophylaxis and as a treatment of infections. The kidney is the main site of metabolism and elimination of the drug [103,104]. Investigators suggest a loading dose of 5 mg/kg, followed by a maintenance dose between 2.5 mg/kg and 5 mg/kg after 18, 24, or 36 h depending on gestational age (>27 weeks or <27 weeks). Authors recommend early monitoring of serum drug concentration to avoid renal toxicity [70,105].

Gentamicin is an aminoglycoside antibiotic characterized by a narrow therapeutic index (risk of nephrotoxicity and ototoxicity). As for other antibiotics of this class, its PK behavior is strongly affected by the patient’s age, body weight, and renal functionality [106]. Different PK studies have suggested, for preterm neonates, a daily dose of 3.5–5 mg based on body weight with longer intervals of 36–48 h [70,107]. Gentamicin is often used in NICUs to treat Gram-negative infections and suspected sepsis [108]. However, in order to limit the risks of trough-associated nephrotoxicity, the use of high doses of gentamicin administered at prolonged dosing intervals has been widely adopted in NICUs in clinical practice [109]. A target trough concentration associated with reduced risks of toxicity for gentamicin is <1 μg/mL, which also minimizes bacterial adaptive resistance thanks to the post-antibiotic effect [110]. However, in vitro studies have suggested, for gentamicin, a PK/PD target of peak concentration (Cmax) over minimum inhibitory concentration (MIC) ratio ranging between 8 and 10 [111]. Recently, Neeli H. and colleagues realized a gentamicin PBPK model developed for preterm and extremely preterm neonates that was evaluated against data collected during clinical practice in a local NICU [98]. Based on their findings, the authors suggest that a higher dose (5 mg/kg), intravenously administered every 36 h, in neonates with a PMA of 30 to 34 and ≥35 weeks is able to minimize the risk of elevated trough concentrations and to provide effective antibacterial activity [98]. A similar conclusion was reached by Valitalo PA and colleagues (2015), who proposed dosing intervals of up to 72 h for both gentamicin and tobramycin, but with a different dose for gentamicin versus tobramycin (4.5 versus 5.5 mg/kg, respectively) [112]. In particular, the authors performed Monte Carlo simulations using validated neonatal pharmacokinetic models of gentamicin and tobramycin in order to evaluate target peak and trough concentration attainment and cumulative AUC over 1 week of treatment [112]. Moreover, they compared the performance of commonly used gentamicin and tobramycin dosing guidelines [113,114,115,116] with the simulated results. In detail, peak concentrations of 5–12 mg/L and trough concentrations <0.5 mg/L were chosen as targets for the proposed dosing guidelines, and the proportion of patients that reached trough concentrations <1 mg/L was calculated. Based on the performed simulations, the proposed dosing guidelines (4.5 mg/kg gentamicin or 5.5 mg/kg tobramycin) with a dosing interval based on birth weight and post-natal age have led to adequate peak concentrations with only 33–38% of the trough concentrations’ target. These novel model-based dosing guidelines have been compared with the simulated performance of the existing neonatal dosing regimens [113,114,115,116]. Simulations based on the existing guidelines revealed adequate peaks but elevated trough concentrations (63%–90% above target) compared to the proposed ones. Therefore, the authors conclude that the proposed neonatal dosing strategies for gentamicin and tobramycin show an improved attainment of target concentrations and should be prospectively evaluated in clinical studies to assess the efficacy and safety of this treatment [112]. The suggested PK/PD target for gentamicin (Cmax/MIC ratio at least 8–10) has been further verified in a cross-sectional observational study with pharmacokinetic analysis performed on both preterm and full-term neonates (totally *n* = 113) [117]. In this study, a weight-based dosing interval (5 mg/kg, q24–48 h) achieved the target gentamicin concentrations for both peak and trough levels in the majority of neonates (*n* = 93/113) [117]. The same dosing interval, but with a slightly higher gentamicin dose (6 mg/kg), was used by Fjalstad JW and colleagues (2013). This high-dose gentamicin (6 mg/kg) regimen has been associated with a low elevated trough plasma concentration (>2 mg/L) and no evidence of ototoxicity [118].

A popPK study on gentamicin in a large cohort of premature and term neonates has confirmed that, compared with term neonates, preterms require longer dosing intervals (up to 48 h), and extremely preterm neonates (below 28 weeks of GA) will also require higher doses of gentamicin (5 mg/kg) to achieve therapeutic concentrations [119]. In particular, these model-based simulations confirmed the high variability of gentamicin kinetics in newborns and that, although PMA was found to be a good predictor of gentamicin CL, the use of covariates such as growth (represented by BW) and maturation (represented by GA and PNA) represents the best approach to describe the gentamicin disposition in preterm neonates [119].

In another study, a PBPK model was developed using the Simcyp Simulator V17 to predict the PK of several drugs, including gentamicin and vancomycin, in preterm neonates [95]. For both gentamicin and vancomycin, the PBPK model prediction for plasma concentration–time profiles after single and multiple intravenous doses has shown a good agreement with the observed data in the preterm population. In terms of physiological parameters, since gentamicin and vancomycin are subjected to kidney elimination, the maturation of the renal function was able to predict the exposure of these two compounds after intravenous administration. Therefore, the authors conclude that, although not substitutive of clinical trials, the prediction of PK behavior in preterm patients using PBPK modeling could be useful to decide on first-time dosing in this population in the absence of clinical data [95].

Aminoglycosides with penicillin are commonly used as a first-line antimicrobial therapy to treat early onset sepsis within different neonatal settings. However, the concerns about emergence of gentamicin resistant Gram-negative organism have led neonatal units to switch to amikacin [120]. Amikacin is used alone as a first-line treatment against Gram-negative bacteria such as *Pseudomonas sp*, *Klebsiella pn*, and *E. Coli*. It is also used in combination with other antibiotics such as vancomycin or meropenem in severe sepsis. Previously, a popPK model in nonlinear mixed-effect modeling software (NONMEM) and a model-based dosing regimen for amikacin in neonates (GA 24–43 weeks, PNA 1–30 days, and birth weight 385–4650 g) were developed [121,122]. In this study, the authors show that birth weight, PNA, and the co-administration of ibuprofen should be preferred to PMA alone as the main covariates for predicting amikacin renal clearance. The authors conclude that a new model-based dosing regimen based on current body weight and PNA could better reflect the maturation of GFR, allowing adjustments of dosing regimens for other renally excreted drugs in preterm and term neonates [121,122]. The model-based dosing regimen proposed by De Cock RF and colleagues (2012) was subsequently evaluated in a prospective study in preterm and term neonates [123]. Specifically, this study was divided into two steps. In the first step, TDM results were evaluated according to predefined target concentrations (trough, <3 mg/L; peak, >24 mg/L) [121,124]. In the second step, the predictive performance of the previously published model was evaluated by comparing the observed TDM results to the model-based predicted concentrations and by analyzing the normalized prediction distribution error (NPDE). Thereafter, Monte Carlo simulations were performed twice in 5000 individuals following the model-based dosing regimen (with or without ibuprofen co-administration) [121]. TDM observations were accurately predicted by the model without bias, which was confirmed by the NPDE. Monte Carlo simulations show that peak concentrations of >24 mg/L were reached at a steady state in almost all patients. Trough values of <3 mg/L at a steady state were documented in 78% to 100% and 45% to 96% of simulated cases with and without ibuprofen co-administration, respectively. Therefore, the authors confirm that a large percentage of neonatal amikacin TDM observations, obtained using a model-based dosing regimen, reached the predefined targets. Finally, they conclude that, although the prospective evaluation of the amikacin dosing regimen may be of relevance beyond the compound-specific observations, TDM remains an indicator to guide safe and effective amikacin therapy. Moreover, for patients with trough levels between 3 and 5 mg/L, which were also confirmed with Monte Carlo simulations after 4 or 5 administrations, the authors suggest a dosing interval of 36 h instead of 30 h and 30 h instead of 24 h to avoid toxicity [123].

In a popPK model developed by Cristea S and colleagues (2017), the impact of therapeutic hypothermia (PATH) on amikacin PK was evaluated in neonates, including preterms, with perinatal asphyxia. In this study, TDM data collected retrospectively from neonates with PATH were combined with previously published data [121] in order to develop a two-compartment model using NONMEM. Birth weight and PNA were assumed to be covariates on CL, whilst current body weight was covariate on the distribution volume, as already reported [121]. Monte Carlo and stochastic simulations were performed to validate the PK model and to establish amikacin exposures in neonates with PATH, comparing current dosing guidelines with the proposed model-derived dosing strategy. PK analysis revealed a significant decrease of amikacin clearance (40.6%) in neonates with PATH. Based on performed simulations, a 15-mg/kg dose every 42 h for children above 2800 g and every 48 h for children between 1800 g and 2800 g has been able to achieve safe trough concentrations (5 mg/L) while still reaching optimal peak concentrations (24 mg/L). Therefore, the authors conclude that a 12 h increase in the amikacin dosing interval in neonates with PATH could be used to correct for the reduced clearance, reaching safe and effective exposures of amikacin [125]. In another study conducted by Claassen K and colleagues (2015), the authors developed a PBPK model of amikacin aiming to predict plasma levels in preterm neonates. In particular, predicted plasma concentration–time profiles, following single and multiple dosing, were compared to experimentally obtained TDM data, revealing an appropriate prediction degree throughout a large range of gestational and postnatal ages. Therefore, the authors conclude that PBPK simulations in preterm neonates appear feasible and might be a useful tool to support dosing decisions in this special patient population [126].

### 6.2. Beta-Lactams

Beta-lactams are the most used antibiotics in newborns. Due to their structural similarity with the enzyme transpeptidase, which is involved in the crosslinking of peptidoglycan chains, beta-lactams act to block the ex novo formation of bacterial cell walls. These antibiotics are time-dependent agents. Therefore, their efficacy is linked to the time spent using the free drug fraction above the MIC value for the pathogen (%fTime>MIC). The optimal PD target should be at least 40–50% of the time, but, in severe infections, 100% of the time is recommended. Beta-lactams have a good safety profile; therefore, TDM is rarely applied to these agents. However, in cases of severe infections, the distribution volume and filtration glomerular rate are significantly increased over time in newborns. These pathological changes dramatically affect the PK of different drugs, including antibiotics. Therefore, in special situations, the monitoring of antibiotics’ concentrations could be particularly advisable. The most used beta-lactams in neonates include penicillins, carbapenems, and cephalosporins.

#### 6.2.1. Penicillins

Among penicillins, ampicillin is the most widely used antibiotic in the neonatal period. It is generally used in association with other antibiotics, mainly aminoglycosides, for both prophylaxis and the therapy of early and late onset infections. As for other antibiotics, ampicillin is almost completely (90%) eliminated by the kidney; therefore, its circulating levels are primarily dependent on renal functionality. Penetration into the cerebrospinal fluid is limited. However, in the case of inflamed meninges, its penetration rises to 39%. Ampicillin is the most used drug in the case of a suspected or confirmed *Listeria meningitis* infection, especially if it occurs in the first week of life [127].

Although ampicillin is one of the most administered antibiotics in NICUs, its PK behavior and safety in neonates are poorly described. Actual dosing regimens take into account the GA- and PMA-related variations in renal drug clearance and recommend lower and less frequent dosing in the most premature neonates [115]. However, the knowledge of ampicillin dosing in the most extremely premature neonates (GA of ≤32 weeks at birth) is still limited. A popPK model developed by Tremoulet A and collaborators (2014) included neonates stratified by GA (<34 or >34 weeks) and PNA (<7 or >7 days). Drug concentrations were used to construct a nonlinear mixed-effects modeling in NONMEM, followed by Monte Carlo simulations, aiming to determine the probability of target attainment for the time in which the total steady-state ampicillin concentrations remained above the MIC (%T>MIC) for 50%, 75%, and 100% of the dosing interval. Results have shown that PMA and serum creatinine are important covariates for ampicillin clearance. Finally, the authors suggest a simplified dosing regimen of 50 mg/kg every 12 h for GA of <34 weeks and PNA of <7 days, 75 mg/kg every 12 h for GA of <34 weeks and PNA of >8 and <28 days, and 50 mg/kg every 8 h for GA of >34 weeks and PNA of <28. They conclude that the revised dosing regimen, based on GA and PNA, is able to achieve the desired therapeutic target in 90% of subjects [128].

Similarly, in a prospective study by Padari H et al. (2021), the authors performed a non-compartmental analysis (NCA) and a popPK modeling on 14 neonates (GA of 32–42 wks) who were receiving ampicillin for suspected or proven early onset sepsis and pneumonia. A visual predictive check for the final PK model was performed to assess the probability of the target attainment of various dosing schemes against MIC of 0.5, 1, 2, 4, and 8 mg/L. As PD targets were evaluated, 40% fT > MIC and 100% fT > MIC, and the safety margin of Cmax > 140 mg/L, assuming 100% and 80% of unbound ampicillin fractions in serum. Based on the simulations’ results, the authors suggest that, during the first week of life in neonates with GA ≥32 weeks, empiric ampicillin dose of 50 mg/kg q12h (iv, intravenous) will achieve plasma concentrations above the group B streptococcal (GBS) breakpoint of 0.25 mg/L susceptibility throughout the dosing interval. In the case of pathogens with a higher susceptibility breakpoint, the dose of 50 mg/kg q8h is sufficient to achieve a target of fT100% > MIC = 2 mg/L. These dosing regimens exceed the safety margin values of Cmax above 140 mg/L in less than a third of patients and, therefore, can be considered sufficiently safe [129].

A retrospective evaluation of previously published popPK models has been performed for both ampicillin and gentamicin on preterm neonates with GA <28 weeks [130]. In particular, Monte Carlo simulations were used to generate concentration–time profiles for the most commonly used dosing regimens of ampicillin 50−100 mg/kg/dose every 8−12 h for 24−48 h courses (i.e., 2–6 doses) and 1 dose of 5 mg/kg gentamicin. The post-discontinuation antibiotic exposure (PDAE), defined as the time from the last dose to the time when the concentration decreased below MIC, was evaluated for both antibiotics. Simulation results show that all ampicillin dosing regimens (50–100 mg/kg every 8–12 h for 2–6 doses) achieved therapeutic exposures > MIC range. After the last dose, the PDAE mean ranged from 34 to 50 h for *E. coli* (MIC = 8) and 82 to 104 h for GBS (MIC = 0.25); the longer PDAE values occurred with higher doses, shorter intervals, and longer courses. Short-course ampicillin (2 doses, 50 mg/kg every 12 h) provided a PDAE of 34 h for *E. coli* and 82 h for GBS. Single-dose 5 mg/kg gentamicin provided PDAE > MIC = 2 for 26 h. The authors conclude that PDAE could be an innovative metric designed to identify opportunities to reevaluate dose–exposure relationships, although prospective studies are necessary to confirm the relationship between PDAE and clinical outcomes [130].

Finally, general recommendations suggest using dosages between 50 and 75 mg/kg every 12 h. In cases where higher dosages (50–100 mg/kg every 8 h) are needed, a close neurological monitoring for the risk of convulsions is advisable, especially for Cmax values above 140 mg/L [128,129,131].

Amoxicillin is a penicillinase-susceptible semi-synthetic amino-penicillin and is a structural and pharmacological relative of ampicillin [132]. Amoxicillin is a time-dependent antibiotic, and the PK/PD index is the fraction of time during which the antibiotic concentration remains above the MIC minimal of the targeted pathogen (%fT > MIC) [133]. The clinical PK of amoxicillin in neonates has been clearly reviewed by Pacifici GM and Allegaert K., (2017) [132]. However, despite the longstanding use of amoxicillin for the treatment of neonatal sepsis, there is a lack of data supporting a tailored dosing strategy. In this context, van Donge T and colleagues (2020) assessed individual intravenous amoxicillin exposures based on six international guidelines, namely, the Swiss Agency for Therapeutic Products 2015 (Swissmedic), the British National Formulary for Children 2015, the Neonatal Formulary 7th edition (NNF7), Frank Shann’s Drug 2014 (Shann), The Harriet Lane 2014, and Lexicomp 2016, by applying a previously developed popPK model [134]. The aim of the study performed by van Donge et al. was to evaluate efficacious and safe exposure for current neonatal amoxicillin dosing regimens using the %fT > MIC as the end points of interest and the potential neurotoxicity with Cmax > 140 mg/L value as the threshold. Exposure was simulated by attributing each dosing regimen to a cohort of neonates with a median (IQR) GA of 34 (29–39) weeks, derived from the clinical data for neonates in the Antibiotic Resistance and Prescribing in European Children (ARPEC) study [135]. Six international guidelines and all surveyed Swiss NICUs provided recommendations for intravenous amoxicillin dosing, ranging from 10 mg/kg every 12 h to 50 mg/kg every 4 h. Total daily doses of amoxicillin in use in Swiss NICUs (50–200 mg/kg/day) were higher than those recommended in the international guidelines (20–200 mg/kg/day) with one exception (Shann; suggesting a maximum total daily dose of 300 mg/kg). Simulation results revealed that none of the dosing regimens achieved targets of ≥100%fT > MIC at any of the relevant MICs for a desired probability of target attainment (PTA) of ≥90%. All dosing regimens achieved a PTA ≥90% for *Streptococcus agalactiae* (MIC 0.25 mg/L) and *Listeria monocytogenes* (MIC 1 mg/L) when targeting ≤ 70%fT > MIC. In contrast, none of the regimens resulted in a PTA ≥ 90% targeting ≥ 70%fT > MIC for *enterococci* (MIC 4 mg/L). In terms of neurotoxicity, the Cmax associated with potential neurotoxicity was exceeded using four dosing regimens (100 mg/kg q12, 60/30 mg/kg q12/8, 50 mg/kg q12/8/6, and 50 mg/kg q12/8/4) for ≥10% of neonates. Therefore, the authors conclude that new randomized trial designs combining both pharmacometric modeling and simulation could be used to select the optimal dosing regimens in preterm neonates [136].

One issue in conducing PK studies in neonates is determined by the limited sampling and the low blood volumes available. A valid approach could be represented using scavenged samples, left over from the routine clinical care of infants without obtaining additional blood. In fact, these samples can be collected in the clinical laboratory from discarded blood (heparinized or EDTA tubes) obtained during routine clinical practice. Therefore, if combined with the collection of timed PK samples (collected ad hoc for study protocols), scavenged samples could be used for PK characterization in preterms [137]. An example of this application was reported by Cohen-Wolkowiez M and colleagues (2012) [68]. These authors developed a popPK model of piperacillin using targeted sparse sampling and scavenged samples obtained from preterm infants ≤32 weeks of gestational age at birth and < 120 postnatal days. This model was developed using nonlinear mixed-effect modeling. Thereafter, Monte Carlo simulations based on the final popPK model were used to explore dose–exposure relationships, adopting the current piperacillin dosing recommendations. From the evaluation of a population’s mean clearance, an augmented CL was observed with the increase of gestational age at birth; newborns with serum creatinine ≥ 1.2 mg/dL show a 60% reduction in piperacillin CL. Therefore, after allometric scaling, serum creatinine was included in CL model, resulting in an increased model fit. The authors conclude that piperacillin dose adjustments will likely be performed considering this parameter. Finally, this study confirms the utility of scavenged sampling in performing PK studies and providing dosing recommendations in preterm neonates. However, this approach is not feasible for unstable drugs, and a compound’s stability is an important covariate that should be considered when using the leftover samples [68].

In another study by the same author, piperacillin-tazobactam PK was evaluated in premature and term neonates of ages <61 days with suspected systemic infections [138]. In particular, neonates were administered intravenous piperacillin-tazobactam (80 to 100 mg/kg of body weight every 8 h) based on GA and PNA. Interestingly, the drugs’ levels were measured in both plasma and dried blood spot collected samples. PK data were analyzed using population nonlinear mixed-effect modeling. The final model was used to generate 1000 Monte Carlo simulation replicates per time point of piperacillin-tazobactam exposure, and the simulated results were compared with those observed in the study. The time unbound piperacillin concentrations remained above the MIC for 50% and 75% of the dosing interval at steady state was evaluated as a target attainment rate. The results obtained for piperacillin and tazobactam PK models show that body weight and PMA are covariates for clearance, whilst body weight is a covariate for the volume of distribution. These covariates were used to optimize dosing in newborns. Moreover, DBS drug concentrations resulted in a 50 to 60% lower amount compared to those in plasma; however, when combined with plasma concentrations, the generated PK model parameters were similar to those for plasma alone. Finally, the authors conclude that, following a PMA-based dosing regimen (100 mg/kg q8h, 80 mg/kg q6h, and 80 mg/kg q4h for PMA of <30, 30 to 35, and 35 to 49 weeks, respectively), 90% of simulated infants were able to achieve the surrogate therapeutic target time above the MIC (<32 mg/L) for 75% of the dosing interval [138].

Piperacillin-tazobactam PK was also evaluated in a popPK model developed by Li Z and colleagues (2013) in moderate preterm newborns (median GA 36.04 weeks). In particular, a total of 207 piperacillin and 204 tazobactam concentration–time datasets from 71 patients were analyzed using a nonlinear mixed-effect modeling approach. Thereafter, the final models were evaluated using both bootstrap and visual predictive checks by simulating one thousand datasets based on the final model. This PK analysis revealed that PMA is the most significant covariate of the central clearance of piperacillin and tazobactam, although the combination of the current body weight and PNA seems to be superior to PMA alone. Moreover, body weight is the most important covariate for the apparent central volume of distribution. Based on these results, the authors conclude that a dosing strategy of piperacillin/tazobactam 44.44/5.56 mg/kg/dose every 8 or 12 h allows researchers to achieve the PD target (free piperacillin concentrations >4 mg/L for more than 50 % of the dosing interval) in about 67% of infants. Finally, the authors suggest that higher doses or more frequent dosing regimens could be necessary for controlling infection in this population in NICU [139].

#### 6.2.2. Carbapenems

Among carbapenems, meropenem is the most used in newborns, especially to treat late-onset sepsis (LOS) and complicated intra-abdominal infections. Its safety profile is good, with rare cases of cytopenia [140,141]. Meropenem is a broad-spectrum antibiotic approved by the US Food and Drug Administration for use in pediatric patients, including treating complicated intra-abdominal infections in infants <3 months of age (MERREM(R) IV (meropenem for injection, summary of the product characteristics) [142]. In neonates, meropenem is currently only approved for treating complicated intra-abdominal infections (cIAIs) sustained by both Gram-positive and Gram-negative bacteria. Whereas meropenem’s pharmacokinetics in adults are well defined, there is a lack of knowledge about PK properties in preterms.

In a recent article by Ganguly S. and colleagues (2021), the authors applied a PBPK modeling to characterize the disposition of meropenem in preterm and term neonates [143]. This model was developed using 645 plasma concentrations from 181 infants (GA 23–40 weeks; PNA 1–95 days). The PBPK-model-based simulations were performed to evaluate suggested meropenem dosing for infants <3 months of age with cIAIs, as reported on the product label. The PBPK-model-predicted clearance in a virtual infant population was successfully able to capture the post hoc estimated clearance of meropenem in this population, as suggested by a previously published popPK model [144]. Similarly, almost 90% of virtual infants showed a 4 mg/L target plasma concentration for >50% of the dosing interval following the product-label-recommended dosing. The authors conclude that the PBPK model supports the meropenem dosing regimens recommended on the product label for infants <3 months of age and that both the PBPK and popPK modeling approaches suggest similar meropenem dosing recommendations for this specific age range [143].

In a study performed by Padari H and colleagues (2012), the authors compared the steady-state PK and safety of meropenem administered at a dose of 20 mg/kg every 12 h via short (30 min) or prolonged (4 h) infusion to neonates with a GA of <32 weeks (birth weight < 1200 g), aiming to define the appropriate dosing regimen for a phase 3 efficacy study of neonatal LOS [145]. Meropenem concentrations were measured immediately before and 0.5, 1.5, 4, 8, and 12 h after the 4th to 7th dose. The results show, for a short infusion, a higher mean drug concentration in serum (Cmax) compared to prolonged infusion (89 vs. 54 mg/L). For intermediate or resistant microorganisms (with meropenem MICs of >2 mg/L), such as *Acinetobacter spp. and Pseudomonas aeruginosa*, previous PK/PD simulation studies involving neonates suggested better PK/PD target attainment with 4 h infusions [146]. In a study by Padari H and colleagues, PK analysis revealed that all the patients in the short-infusion group and 8/10 in the long-infusion group achieved the PD target %fT>MIC of 100% for a MIC of 2 mg/L. Moreover, the meropenem clearance was not influenced by postnatal or postmenstrual age, and the one-compartment popPK model demonstrated that covariates serum creatinine, PNA and GA, were not able to improve the best model fit. Based on these results, the authors conclude that the final parameters estimated are the steady-state distribution volume (Vss) of 0.301 L/kg and the CL of 0.061 L/h/kg. Moreover, at a MIC cut-off of 8 mg/L with a short infusion, no neonate is expected to have a %fT>MIC of 40%, with target values of >95%. Therefore, they conclude that, in very-low-birth-weight neonates, meropenem infusions of 30 min are optimal, since they guarantee a reasonable balance between high Cmax and %fT>MIC for susceptible organisms with no dosing adjustments over the first month of life. Additionally, they suggest that a dose of 20 mg/kg given as a 30 min infusion could be used in a larger study of efficacy in patients with LOS [147]. Conversely, a prospective, randomized clinical trial compared the intravenous infusion of meropenem over 4 h (infusion group) or 30 min (conventional group) at a dosing regimen of 20 mg/kg/dose every 8 h and 40 mg/kg/ dose every 8 h in neonates (GA 33–34 weeks) with Gram-negative late-onset sepsis (GN-LOS) admitted to NICU [148]. The results of this study revealed that the prolonged infusion of meropenem is better associated with clinical improvement, microbiologic eradication, and less neonatal mortality compared to the conventional strategy [148].

Doripenem is a parenteral carbapenem with broad-spectrum activity against aerobic Gram-negative and Gram-positive pathogens and anaerobic pathogens. The PK behavior, safety, and tolerability of doripenem were evaluated in a phase I study that also included neonates with chronological ages (CA) less than 4 weeks (<32 weeks and ≥32 to ≤44 weeks in GA) [149]. The results show that a single dose of doripenem (5 mg/kg of body weight for <8 weeks and 8 mg/kg for >8 weeks in chronological age) administered as a 1 h infusion in term and preterm newborns <12 weeks CA was similar to what was observed in neonates and very young infants with other carbapenems (PK/PD target attainment %T >MIC between 70–99%) [149].

Imipenem is a broad-spectrum antibacterial agent used in critically ill neonates after failure of first-line treatments [150]. A recent popPK analysis developed by Dao K and colleagues (2021) including preterm neonates with a median GA of 27 weeks (range: 24–41). PK data were analyzed using a one-compartment non-linear mixed-effect modeling and revealed that GA and PNA exhibited the greatest impact on the PK parameters, followed by serum creatinine. Moreover, simulations using a dosing regimen of 20–25 mg/kg every 6–12 h according to PNA led to the highest percentage of target attainment (T>MIC over 100% of time). Therefore, the authors conclude that a dosing adjustment according to body weight, GA and PNA is the best strategy to optimize imipenem exposure in neonates [151].

#### 6.2.3. Cephalosporins

Along with a large total-body water volume and immature renal function, neonates are also characterized by low albumin levels [46]. This last aspect should be considered when administering drugs with a high percentage of protein-bond. Cefazolin is highly bound to human serum albumin, and its indications in neonates are mainly prophylactic (72%) and, to a lesser extent, therapeutic (17%) (e.g., coagulase-negative staphylococcal sepsis) [152]. Since exclusively unbound cefazolin distributes to the extravascular compartments and is subjected to renal elimination, low albumin levels could affect cefazolin disposition. In this context, a popPK model was realized using both total and unbound cefazolin plasma concentrations as a guide for dosing in preterm and term neonates [153]. In this study, the popPK analysis was performed on 119 total and unbound plasma concentrations of cefazolin obtained from 36 (pre)term neonates with PNA 1–30 days. Monte Carlo simulations were applied, aiming for unbound concentrations above a MIC value of 8 mg/L (60% of the time) in all patients. The results of this one-compartment PK model show that the current BW is the main covariate for Vd, whereas birth BW and PNA are the main covariates for clearance and albumin plasma concentrations for maximum protein binding (Bmax). Moreover, based on simulations, the authors proposed a body-weight- and PNA-adapted dosing regimen that resulted in similar exposure across different weight and age groups. Finally, the authors conclude that both the total and unbound cefazolin concentrations in neonates can be described in a one-compartment popPK model that includes saturable protein binding. Moreover, birth BW and PNA are considered the main covariates affecting the variability in cefazolin CL. Therefore, they propose a new model-based neonatal cefazolin dosing regimen suggesting, however, a future prospective validation of their model [153].

Cefotaxime is another antibiotic widely prescribed to treat Gram-negative bacterial sepsis in neonates [154]. However, dosing regimens are often characterized by high variability rates [155]. A popPK model was developed by Leroux et al. (2016) by elaborating data from 100 neonates (GA range 23–42 weeks) with an allometric two-compartment model. This PK analysis indicated the current weight, GA, and PNA as significant covariates. Monte Carlo simulations have been used as visual prediction validation of the PK model aiming to assess a target attainment of fT>MIC of 75% of the dosing interval at steady state for each dosing regimens. Based on this model validation, the authors proposed a dosing regimen of 50 mg/kg between and four times a day, according to GA and PNA, in order to improve dosing in older newborns (PNA > 1 week and/or GA > 32 weeks, time > MIC 75%) [156].

Ceftolozane/tazobactam is a combination of the β-lactam/β-lactamase inhibitors that has a broad-spectrum activity against the most common Gram-negative bacteria, including MDR strains. The PK and safety profile of this combination was evaluated in a phase I, noncomparative, open-label, multicenter study on pediatric patients with proven/suspected Gram-negative infections or perioperative prophylaxis receiving a single intravenous (iv) dose of ceftolozane/tazobactam [157]. In particular, patients were divided into two groups: Group A (GA > 32 weeks) and Group B (GA ≤ 32 weeks). The results show that PK profiles in neonates and young infants were generally comparable to those of older children receiving a single iv dose of ceftolozane/tazobactam. Therefore, the authors conclude that, among term and premature neonates and young infants, PK was comparable to older children, and that ceftolozane/tazobactam was generally well tolerated. However, they highlight the necessity of proper neonatal PK trials [157].

Ceftazidime (CAZ) belongs to third-generation cephalosporins. It is approved for children > 1 month. Its use in neonatal age is limited to severe infections, especially with cerebro-spinal fluid (CSF) involvement. Old studies in preterm demonstrated that the clearance of CAZ increases with gestational age and higher GFR [158]; in the first 10 days of life, the GFR is increased, and the CAZ clearance is consequently accelerated, whereas the distribution volume and elimination half-life are significantly reduced between day 3 and day 10 after birth. The dosage of CAZ is 25–50 mg/kg bw twice daily, but attention must be paid to concomitant medications that reduce GFR, such as indomethacin [159,160]. The emergence of extensively drug-resistant (XDR) or pan drug-resistant (PDR) Gram-negative bacteria is also a major concern in NICU. Treatment options are limited, and mortality is high. Complicated abdominal infections and severe sepsis are the most frequent manifestations. Ceftazidime/avibactam (CAZAVI) is a novel combination of ceftazidime with a new beta-lactamase inhibitor, avibactam, a non-beta-lactam/beta-lactamase inhibitor with good activity against XDR *Enterobacteriaceae* (including *Klebsiella pneumoniae* carbapenemase producer), *Pseudomonas aeruginosa and Acinetobacter baumanii*. CAZAVI is licensed for use in infants > 3 months [161]. PK behavior in children was already evaluated in a phase I study and two phase II studies [162,163]. A dose of 10–40 mg/kg q8h for those ≥ 3 to 6 months old with creatinine clearance > 50 mL/min/1.73 m^2^ was suggested. The safety results were similar to ceftazidime alone, and treatment appeared effective in pediatric patients with complicated abdominal infections [162]. In terms of CAZAVI use in preterms, a report was published showing similar results related to the efficacy and safety profile [164]. However, future prospective trials on this population are needed.

### 6.3. Glycopeptides

Glycopeptides inhibit the synthesis of cell-well peptidoglycan and affect bacterial cell membrane permeability [165] Oritavancin, a newer lipoglycopeptide derivative, may also have an effect on the inhibition of RNA synthesis, but a direct involvement is still debated [166]. They are used to treat severe infections sustained by Gram-positive bacteria, including methicillin-resistant *Staphylococcus aureus* and coagulase negative staphylococci (CoNS), which are often the cause of late-onset sepsis in newborns [166].

#### 6.3.1. Vancomycin

Preterms or neonates born with very low birth weights are particularly susceptible to serious Gram-positive infections during their NICU stay. In fact, the use of central venous catheters and total parenteral nutrition is often a source of infections. This augmented susceptibility is also due to an immature immune system that is not able to guarantee an adequate anti-microbial response. *Staphylococcus aureus* and coagulase-pathonegative staphylococci are responsible for almost 55% of late-onset nosocomial infections in newborns [167]. The glycopeptide antibiotic, vancomycin, is widely used to treat methicillin-resistant *S. aureus* infections in premature and full-term neonates [168]. Vancomycin is water-soluble, has a limited plasma protein binding capacity (i.e., IgA and albumin), and is mainly eliminated by the kidneys via glomerular filtration and renal tubular transport. Compared to adults, neonates have a higher extracellular fluid volume and a limited renal elimination rate. Most premature neonates present with a higher distribution volume and low renal clearance; therefore, vancomycin clearance may vary 2-to 3-fold according to the neonatal age range and co-morbidity [169,170,171]. Consequently, vancomycin dosing is different based on either PMA or serum creatinine levels. In this context, TDM and individualized treatments should be warranted in neonates treated with vancomycin [172,173].

Similar to adults, continuous vancomycin infusion has been used in newborns. The advantages are represented by better target attainment and an easier interpretation of drug levels. However, the ideal dosing regimen for the administration of vancomycin in neonates is still debated [174,175]. However, continuous infusion does not exclude the influence that fluid status and comorbidities exert on vancomycin clearance, especially in critically ill patients [176].

So far, several dosing algorithms have been proposed and used during routine clinical practice [177]. These algorithms include a fixed dose based on body weight for all neonates, PMA-based dosing, PMA- and weight-based dosing, PMA- and PNA-based dosing, and serum-creatinine-based dosing [178]. These four common dosing regimens for vancomycin in preterm and term neonates were compared using a popPK model followed by Monte Carlo simulations in order to assess the probability that each regimen would achieve the widely used therapeutic target serum trough concentrations of 5–15 mg/L and the newly suggested target for methicillin-resistant *S. aureus*, of 15–20 mg/L [177,179]. In a study by Mehrotra N. et al. (2012) [178], TDM data for vancomycin were collected from 134 preterm (66%) and term (34%) neonates, with a PNA of 1–121 days and PMA of 24.6–44 weeks. These data were used to develop a popPK model in this target population followed by Monte Carlo simulations for four recommended dosing regimens: a standard dose for all neonates, PMA-based dosing, PMA and PNA-based dosing, and serum-creatinine-based dosing. The results obtained from these comparisons demonstrated that serum-creatinine-based dosing shows the highest chance of reaching the target trough concentration range of 5–15 mg/L. Therefore, the authors conclude that, although this may sometimes be challenging in the neonatal setting, measuring the serum creatinine concentration before dosing vancomycin in preterms could be useful to reach therapeutic drug concentrations [178].

The most recognized PK/PD target for vancomycin is the 24 h area under the concentration–time curve (AUC 0–24)-to-MIC ratio (AUC 0–24/MIC) of >400 for microorganisms with a MIC value up to 1 mg/L [179,180,181,182], although some authors have supposed that a lower target may be effective in neonates [183]. However, this PK/PD target was originally defined in adult methicillin-resistant *Staphylococcus aureus* (MRSA) pneumonia [181] and was never validated in neonatal Staphylococci septicemia [184].

Conversely, the above PK/PD target was further validated in a popPK “meta-model” performed by Jacqz-Aigrain E and colleagues (2019) [185]. In this study, a “meta-model” was built using NONMEM with vancomycin concentrations from 1631 neonates (median GA ranging from 22.3–42.1), and Monte Carlo simulations were performed to design an optimal intermittent infusion, aiming to reach a target AUC 0–24 of 400 mg*h/L at steady-state in at least 80% of neonates [185]. The results of this PK analysis indicated current body weight, PMA and serum creatinine to be significant covariates for vancomycin CL. After model validation, simulations show that a loading dose (25 mg/kg) and a maintenance dose (15 mg/kg q12h or 15 mg/kg q8h based on PMA) were able to reach the AUC 0–24 target earlier than the suggested “Blue Book” dosage regimen [186] in >89% of the treated patients. Therefore, the authors suggest that this dosing regimen could be used for neonates and to assist in the design of a model-based, multinational, European trial, named NeoVanc [185]. It is also worth noting that the AUC 0–24/MIC target level of 400 is based on the total vancomycin concentration, whilst Smits et al. recently demonstrated that the fraction unbound (FU) of vancomycin is much higher in neonates (median 0.9) compared with adults (median 0.6) [187]. In this study, the authors claimed that the traditional “total drug target approach” is aimed at achieving similar total vancomycin exposure in neonates as in adults without considering the differences in protein binding and thus targeting a common PK/PD index of AUC 0–24/ MIC ≥ 400 for optimal dosing in neonates. Therefore, these authors proposed a novel “unbound drug target approach” aimed at achieving similar unbound vancomycin exposure in neonates as in adults and thus considering an AUC 0–24/MIC ≥ 267 for optimal dosing in neonates to be a target [187]. Based on this finding, Leroux S and colleagues (2019) [184] evaluated the impact of this “unbound drug target approach” on vancomycin dosing by using a PK/PD simulation of 249 preterm neonates that were enrolled in a previous pharmacokinetic study [63]. Specifically, in this neonatal population, the overall medians (ranges) GA, PNA, and BW were 29 weeks (23–34), 11 days (1–27), and 1200 g (415–2630), respectively. These neonates received vancomycin in a 60 min infusion at a dose of 15 mg/kg once or twice a day according to their postnatal age and serum creatinine value [63]. The vancomycin PK parameters and exposure profiles of these 249 neonates were analyzed using a PK model previously developed during a popPK meta-analysis of vancomycin in neonates [185]. In the selected cohort of neonates, the population PK parameters obtained with this PK model [185] produced a vancomycin AUC 0–24 values (mean ± SD) of 446.4 ± 161.3 mg∙h/L at a steady state. Based on the traditional “total drug target approach,” the AUC 0–24/MIC ≥ 400 target was achieved by only 54.2% of the neonates (for a MIC of 1 mg/L). Meanwhile, with the “unbound drug target approach,” the AUC 0–24/MIC ≥ 267 target was reached by 91.2% of the neonates (for a MIC of 1 mg/L) [184]. Thereafter, in order to assess how this new finding based on the “unbound drug target approach” will guide the optimal use of vancomycin as an intermittent infusion in preterm neonates, the authors performed Monte Carlo simulations (*n* = 100) for different dosing regimens. Considering the “unbound drug target approach,” a dosing regimen of 10 mg/kg BID for neonates with a PMA of less than 30 weeks and 10 mg/kg TID for neonates with a PMA of 30 weeks or more was sufficient to achieve a 90% probability of target attainment (AUC 0–24/MIC ≥ 267) at a steady state. Moreover, the vancomycin trough concentrations associated with this dosing regimen were 12.7 mg/L (5th to 95th percentile: 5.1–26.5) at a steady state. Therefore, the authors conclude that, considering the maturational changes in vancomycin protein binding, it should not be feasible to consider a similar AUC 0–24/MIC target level for vancomycin in both neonates and adults. Consequently, they suggest also considering the impact of a higher unbound fraction in neonates when administering vancomycin to these patients [184]. However, considering the well-known limitations in calculating the AUC 0–24 in neonates, the trough concentration is more routinely applied in clinical practice for drug monitoring. In this context, a one-compartment popPK model was developed by Frymoyer A and colleagues (2014) [188] to examine the relationships between troughs and AUC 0–24 in neonates. In terms of covariates, the clearance (CL) was predicted by BW (an indicator of size), PMA (an indicator of maturation), and serum creatinine (Cr; an indicator of renal function). Monte Carlo simulations were performed to assess the effect of dose, PMA, and serum creatinine level on troughs and AUC 0–24 achievements. Based on their results, the authors conclude that a target vancomycin trough concentration between 7 and 11 mg/L is highly predictive of an AUC 0–24 of >400 across simulated neonates for various PMAs, serum creatinine (Cr) levels, and dosing strategies. Moreover, they suggest that higher trough concentrations of 15 to 20 mg/L, as usually recommended in adults, are unnecessary in neonates based on AUC 0–24/MIC when treating neonates for invasive MRSA infections with an MIC of ≤1 mg/L [188]. This model, developed by Frymoyer A and colleagues (2014), was retrospectively validated on a cohort of 243 neonates with a median GA of 30 weeks (range: 22–41) and a median weight of 1.6 Kg (range: 0.4–6.8) [189]. The aim of this study was to conduct an external evaluation of the published pharmacokinetic model and to confirm the relationship between the vancomycin trough concentration and AUC 0–24 in neonates. The results of this study show that the model was able to predict the observed vancomycin concentrations with reasonable precision. Moreover, these data further confirm that in neonates a vancomycin trough concentration of 15–20 mg/L is unnecessary to achieve an AUC 0–24/MIC ≥400 with a MIC ≤1 mg/L and that lower trough concentrations (approximately 10 mg/L) are likely adequate to provide adequate exposure for invasive MRSA while also appropriately covering for coagulase negative staphylococcal infections [189].

The clinical utility and safety of model-based dosing regimens for vancomycin were evaluated in a study performed by Leroux S et al. (2016) [190]. In particular, the authors applied a model-based vancomycin dosing calculator, developed from a previously published popPK model [191], to the routine clinical care in three neonatal intensive care units. This model-based application of vancomycin dosing was demonstrated in 190 neonates with a mean GA and a mean PNA of 31.1 weeks and 16.7 days, respectively. The percentage of patients with a first-serum vancomycin concentration achieving the target window of 15 to 25 mg/L was selected as the endpoint for evaluating the clinical utility. The model-based dosing regimen (determined by birth weight, current body weight, PNA, and serum creatinine) was based on a loading dose of 11.1 mg/kg/day infused over 60 min and followed by the maintenance dose of 28.3 mg/kg/day administered as a continuous infusion over 24 h. The safety evaluation was focused on nephrotoxicity, which was evaluated based on changes in serum creatinine concentrations from the baseline obtained within 48 h of starting the vancomycin administration. The results obtained from this PK model application to clinical practice reveal that the target attainment rate increased from 41% to 72% of neonates without any case of vancomycin-related nephrotoxicity. However, the authors conclude that a prospective controlled trial is needed to further confirm their data [190].

An external validation of a previously published popPK model for vancomycin was performed by Janssen E et al. (2016) [192]. In particular, the aim of this study was to evaluate the predictive performance of the previously published neonatal and pediatric pharmacokinetic models [74,122] against an external vancomycin dataset containing TDM data from both preterm (median GA of 32 weeks) and term neonates and infants [193]. The model used for this study was previously developed by De Cock RFW and colleagues (2014), who proposed a semi-physiological function for the GFR-mediated clearance used to establish evidence-based dosing regimens of renally excreted antibiotics, including gentamicin, tobramycin, and vancomycin [74,122]. For its external validation, the previously published popPK models were used to simulate each of the observations of the datasets 1000 times. Concentration–time profiles were simulated in neonates and children for different dosing regimens reported in the Dutch Children’s Formulary [113], British National Formulary for Children (BNFc) [116], the regimen proposed by the Infectious Diseases Society of America (IDSA) [180], and the meningitis regimen of the NeoFax manual [115], using the parameter estimates from the original models. These simulations were performed in order to evaluate current dosing regimens and to propose a model-based dosing algorithm. A PK/PD target AUC 0–24/MIC > 400 without any concentration exceeding 40 mg/L was evaluated for each simulated dosing regimen. The results show that both the neonatal and pediatric models were able to describe the observed data in the external dataset well. However, with the currently used dosing regimens, the target AUC 0–24/MIC and trough concentrations were hardly reached in neonates and young infants. Therefore, the authors proposed a dosing algorithm based on body weight at birth and PNA for neonates, with daily doses divided over three to four doses. In particular, for infants aged <1 year, doses between 32 and 60 mg/kg/day over four doses are proposed, while above 1 year of age, 60 mg/kg/day seems appropriate. Moreover, in order to reach an AUC 0–24/MIC of 400 on the first day of treatment, a loading dose should be administered. Finally, the authors conclude that a prospective clinical study should be performed to validate this model-based dosing algorithm [192].

Similarly, a model-based dosing approach designed to individualize empiric vancomycin dosing in neonates was retrospectively applied to data from 492 neonates (median GA 32 weeks, range 24–42) treated with vancomycin in two healthcare systems, and empiric dose recommendations from the following four sources were examined: Neo-Vanco, Neofax [194], Red Book [195], and Lexicomp [196,197]. Predicted AUC 0–24 and troughs concentrations were also calculated and compared. Neo-Vanco was developed based on a published, externally validated population pharmacokinetic model that incorporates predictors of PMA, weight, and serum creatinine level [189,198]. Using a simulation-based methodology, an individualized dose aimed at attaining an AUC 0–24/MIC ratio of >400, while reducing trough concentrations of >20 mg/L (toxicity target), was calculated. The final aim of the study conducted by Frymoyer A and colleagues was to compare expected vancomycin exposure levels in neonates on the basis of a Neo-Vanco-derived dosing strategy to those from three commonly used recommendations, Neofax [194], Red Book [195], and Lexicomp [196]. The results show that the percentage of neonates predicted to achieve an AUC 0–24/MIC of >400 was 94% with Neo-Vanco, 18% with Neofax, 23% with Red Book, and 55% with Lexicomp (all *p* < 0.0001 vs. Neo-Vanco). Meanwhile, the predicted troughs of >20 mg/L were inconstant and similar across the dosing approaches. Therefore, the authors conclude that this model-based approach to individualizing empiric vancomycin doses in neonates was able to improve the achievement of target exposure levels and can be easily adopted in clinical practice based on easily available clinical characteristics (weight, PMA, and serum creatinine level). However, a future prospective validation is required [197].

In a preterm pilot study, the authors used data from eight preterm infants with neonatal ventriculitis (median GA 25.3 weeks; range 23.9–27.7) treated with intraventricular vancomycin at a standard starting dose of 15 mg/kg, in order to develop a popPK model on the use of intraventricular vancomycin in the preterm population [199]. Three covariates (serum creatinine, ventricular index (VI), and CSF protein) were tested on the model, whilst the AUC and average CSF concentration predictions were generated from the final model. Time to sterilization, defined as the length of time taken for CSF WCC (white cell count) to fall to <20/mm^3^ and simultaneously achieve sterile CSF, was considered to be a PD target. The results show that covariates of VI and the CSF protein did not demonstrate any influence on CSF vancomycin and that time to sterilization with higher CSF AUC (0–24) and average concentration tends to be shorter. The authors conclude that further study with a larger data pool will be necessary to investigate the influence of VI on CSF vancomycin and to optimize the best dosing strategy [199].

#### 6.3.2. Teicoplanin

Among glycopeptides, teicoplanin has bactericidal activity and efficacy against Gram-positive bacteria, such as methicillin-resistant staphylococci, including coagulase-negative staphylococci (CoNS), which is comparable to that of vancomycin [200]. Although they share a similar mechanism of action, the teicoplanin PK properties are different from vancomycin. In fact, whilst protein bonds are between 30 and 55 % for vancomycin, teicoplanin is highly bound to serum albumin (90%), resulting in a half-life that ranges from 100 to 170 h compared to 6–12 h for vancomycin [201,202]. Therefore, it can be administered once daily either intravenously or intramuscularly. Moreover, it is worth noting that a lower incidence of adverse events, including that of nephrotoxicity, has been reported for teicoplanin compared to vancomycin [203,204]. Therefore, teicoplanin has become one of the most prescribed antibiotics by neonatologists in NICUs [205].

A loading dose of 16 mg/kg at day 1, followed by a maintenance dose of 8 mg/kg daily is considered the gold standard to achieve the optimal efficacy with a targeted trough concentrations (Ctrough) > 10–30 mg/L depending on the severity of infection. Nevertheless, limited data exist in terms of teicoplanin PK/PD properties in neonates, and there is growing evidence that teicoplanin PK displays considerable variability in children in comparison to adults, suggesting the application of TDM in routine clinical practice [206,207,208]. Moreover, the PK/PD target that better correlates with teicoplanin in vitro activity is the Ctrough with an ideal value ≥ 10 mg/L [209]. However, this value could be variable according to the site of infection (Targocid^®^, summary of product characteristics) [210].

In order to assess the optimal dosing regimen, especially in preterm newborns, a popPK model was developed by Kontou A and colleagues (2020) [211]. In particular, the authors analyzed plasma teicoplanin concentrations from 60 neonates with PMAs of 26 to 43 weeks using a nonlinear mixed-effects modeling approach to develop a popPK model with NONMEM software. Monte Carlo simulations were performed to evaluate currently recommended dosing (a loading dose of 16 mg/kg and a maintenance dose of 8 mg/kg/day) using a PK/PD index and the AUC/MIC ratio of ≥400 based on vancomycin experience. The results of this study show that teicoplanin PK is variable in neonates and that body weight is the most significant covariate affecting PK parameters, while the estimated creatinine clearance is also an important covariate on teicoplanin CL. Moreover, the Monte Carlo simulation demonstrated that, with the current dosing regimen, an AUC/MIC ratio of ≥400 was reached by only 68.5% of neonates with a current body weight of < 1 kg when the MIC was equal to 1 mg/kg, versus 82.2%, 89.7%, and 92.7% of neonates with body weight of 1 to <2, 2 to <3, or ≥3 kg, respectively. Therefore, the authors conclude that the current teicoplanin dosing regimen is not suitable for preterm neonates with extremely low birth weights (ELBW) and those with body weight < 2 kg. Additionally, based on their simulations, a stratification of doses according to body weight minimizes the number of patients with suboptimal teicoplanin exposures. In fact, neonates with a body weight < 2 kg may need a higher maintenance dose than the 8 mg/kg currently recommended for pathogens with MIC values of ≤ 1 mg/L, while, for neonates with a body weight ≥ 2 kg, the recommended doses seem to be adequate. An increase of the maintenance dose up to 10 mg/kg and 11 mg/kg for preterm neonates with a BW of 1 to <2 kg and <1 kg, respectively, can help to reach the therapeutic targets early in therapy and reduce the risk of therapeutic failures [211].

### 6.4. Fluoroquinolones

An application of TDM for studying the PK of antibiotics in preterm neonates was reported in three interesting case reports of preterm newborns affected by *Mycoplasma hominis* meningitis and treated with moxifloxacin [212,213,214]. In particular, in the case report described by Yeung T and colleagues, an extremely preterm male (GA = 25 weeks) was treated with doxycycline (4 mg/kg IV every 24 h) and moxifloxacin (5 mg/kg IV every 24 h). TDM was applied to measure the serum concentrations of moxifloxacin and to estimate the pharmacokinetic and pharmacodynamic parameters. These parameters were compared to the targets described in other case reports of *M. hominis* meningitis. In particular, Cmax was 2.5 mg/L whilst the AUC was 28.1 mg∙h/L. Considering the MIC values reported in the literature, the estimated Cmax/MIC for this patient was 21 to 158 (target Cmax/MIC: >10), and the AUC/MIC was 234 to 1757 (target AUC/MIC: ≥100). This report describes the successful treatment of *M. hominis* neonatal meningitis and provides important information on the PK/PD parameters of moxifloxacin in preterm neonates. Moreover, it highlights the importance of performing TDM in order to monitor the target attainment rate [214].

**Table 2 antibiotics-11-01142-t002:** Antibiotics’ PK studies conducted on preterm newborns.

Antibiotic	GA ^1^ (Range)	PK Model	Type of Study	PK/PD Target Evaluated	Suggested Dosing Strategy	Reference
Meropenem	23–40	PBPK	Developed ex novo	50% T>MIC for a MIC = 4 mg/L and 75% T > MIC for a MIC = 2 mg/L	PBPK model supports the meropenem dosing regimens recommended in the product label for infants <3 months of age (MERREM(R) IV)	[143]
Meropenem	<32	popPK	Developed ex novo	100% fT>MIC for a MIC = 2 mg/L	20 mg/kg given as a 30-min infusion	[147]
Imipenem	24–41	popPK	Developed ex novo	100% of the T>MIC	20–25 mg/kg every 6–12 h according to PNA	[151]
Gentamicin	30–34 *	PBPK	Developed ex novo	Ctrough < 1 μg/mL to reduce risks of toxicity	5 mg/kg intravenously administered every 36 h in neonates with a PMA of 30 to 34 and ≥35 weeks	[98]
Gentamicin and tobramycin	0–27 days **	popPK	Validation of previously published models	Cmax concentrations of 5–12 mg/L and Ctrough < 0.5 mg/L	4.5 mg/kg for gentamicin and 5.5 mg/kg for tobramycin intravenously administered every 72 h	[112]
Gentamicin	≤28 to 36	/	Cross-sectional observational study	Cmax/MIC ratio at least 8–10	5 mg/kg, q24–48 h	[117]
Amikacin	≤24 to 41	/	Prospective evaluation of a model-ased dosing regimen	Ctrough, <3 mg/L; Cmax, >24 mg/L	15-mg/kg every 36 h instead of 30 h and 30 h instead of 24 h according to PNA	[123]
Ampicillin	<34 or >34	popPK	Developed ex novo	T>MIC for 50%, 75%, and 100% of the dosing interval	50 mg/kg every 12 h for GA of <34 weeks and PNA of <7 days, 75 mg/kg every 12 h for GA of <34 weeks and PNA of >8 and <28 days, and 50 mg/kg every 8 h for GA of >34 weeks and PNA of <28	[128]
Ampicillin	32–42	NCA and popPK	Developed ex novo	40% fT > MIC; 100% fT > MIC and the safety margin of Cmax > 140 mg/L assuming100% and 80% of unbound ampicillin fractions	In neonates with GA ≥32 weeks, iv dose of 50 mg/kg q12h or q8h in case of pathogens with higher susceptibility breakpoint	[129]
Ampicillin and gentamicin	22–27	/	Validation of previously published models	Post-discontinuation antibiotic exposure (PDAE) for specific MIC values	Short-course ampicillin (2 doses, 50 mg/kg every 12 h) have provided a PDAE of 34 h for *E. coli* and 82 h for group B streptococcal (GBS). Single-dose 5 mg/kg gentamicin provided PDAE > MIC = 2 for 26 h	[130]
Amoxicillin	29–39	/	Comparison between different dosing regimens by using Monte Carlo simulations	100% fT >MIC for efficacy Cmax > 140 mg/L value as threshold for safety	Simulation results have revealed that none of the dosing regimens guidelines have achieved targets of ≥100%fT >MIC at any of the relevant MICs for a desired probability of target attainment (PTA) of ≥90%	[136]
Piperacillin-tazobactam	<61 days **	popPK	Developed ex novo	50% and 75% T >MIC (considering unbound piperacillin concentrations)	100 mg/kg q 8 h, 80 mg/kg q 6 h, and 80 mg/kg q 4 h for PMA of <30, 30 to 35, and 35 to 49 weeks, respectively	[138]
Piperacillin-tazobactam	median 36.04	popPK	Developed ex novo	Unbound piperacillin concentrations >4 mg/L for more than 50 % of the dosing interval	Piperacillin/tazobactam 44.44/5.56 mg/kg/dose every 8 or 12 h allows researchers to achieve the PD target in about 67% of infants	[139]
Cefazolin	1–30 days **	popPK	Developed ex novo	60% T >MIC (for unbound cefazolin concentrations > MIC = 8 mg/L)	25–50 mg/kg every 8 or 12 h according to PNA and current body weight	[153]
Cefotaxime	23–42	popPK	Developed ex novo	fT>MIC for 75% of the dosing interval	50 mg/kg twice a day to four times a day, according to GA and PNA	[156]
Vancomycin	24.6–44 *	popPK	Developed ex novo	serum Ctrough 5–15 mg/L or 15–20 mg/L for methicillin-resistant *S. aureus*	Serum-creatinine-based dosing shows the highest chance of reaching the target trough concentrations range of 5–15 mg/L	[178]
Vancomycin	22.3–42.1	popPK	Developed *ex novo*	AUC 0–24 of 400 mg*h/L at steady-state	A loading dose (25 mg/kg) and a maintenance dose (15 mg/kg q12h or 15 mg/kg q8h based on PMA)	[185]
Vancomycin	23–34	/	Validation of previously published model	AUC 0–24/MIC ≥ 267 at steady state (considering unbound vancomycin concentrations	10 mg/kg BID for neonates with a PMA of less than 30 weeks and 10 mg/kg TID for neonates with a PMA of 30 weeks	[184]
Vancomycin	median 30.0	/	Validation of previously published model	Trough concentrations (approximately 10 mg/L) are predictive of an AUC 0–24/MIC ≥400	15 mg/kg every 6, 8, or 12 h and 20 mg/kg every 12 or 24 h according to PMA and serum creatinine level	[189]
Vancomycin	mean 31.1	/	Validation of previously published model	serum Ctrough 15–25 mg/L	A loading dose of 11.1 mg/kg/day infused over 60 min followed by the maintenance dose of 28.3 mg/kg/day administered as a continuous infusion over 24 h	[190]
Vancomycin	median 32.0	/	Validation of previously published models	AUC 0–24/MIC > 400	For infants aged <1 year, doses between 32 and 60 mg/kg/day over four doses, while above 1 year of age, 60 mg/kg/day seems appropriate	[192]
Vancomycin	24–42	/	Retrospective evaluation of different model-based dosing approaches	AUC 0–24/MIC ratio of >400, while reducing trough concentrations of >20 mg/L (toxicity target)	Neo-Vanco derived dosing strategy based on weight, PMA, and serum creatinine level	[197]
Teicoplanin	26–43 *	popPK	Developed *ex novo*	AUC/MIC ratio of ≥ 400	A loading dose of 16 mg/kg and a maintenance dose up to 10 mg/kg and 11 mg/kg for preterm neonates with a BW of 1 to <2 kg and <1 kg, respectively	[211]
Moxifloxacin	25	/	Case report	Cmax/MIC: >10 AUC 0–24/MIC: ≥100	5 mg/kg iv every 24 h	[214]

^1^ Expressed in weeks; * Post-menstrual age (PMA); ** Post-natal age (PNA).

## 7. Therapeutic Drug Monitoring (TDM) of Antibiotics in Preterms

TDM is used to identify the correct dose for each patient. In fact, the measurement of a drug’s concentrations provides information about the extent of the patient’s exposure to that medication [215]. TDM can also be considered a useful tool for the attainment of antimicrobial stewardship, allowing researchers to achieve optimal clinical outcomes, minimize toxicity, reduce costs, and limit antibiotics resistance. In preterms, antibiotic stewardship could have a multidisciplinary approach. In fact, these particular patients are characterized by numerous variables in terms of PK and Pharmacogenetic (PG), making the evaluation of PK behavior not easily predictable [216].

An antibiotic’s concentration in plasma or in other biologic fluids can be determined by different analytical techniques: immunoassay (IA), high performance liquid chromatography coupled by fluorescence detector (FLD), diode array detector (DAD) or variable wavelength detector (UV), and mass spectrometry (MS) associated with liquid chromatography (LC) or gas chromatography (GC).

Immunoassays, based on the principle of antibody–antigen reactions, can identify and quantify antibiotics in plasma and in other biological matrices. IA allows the easy implementation, automation of systems, daily performance, and reproducibility between laboratories. Validated kits are also available for the determination of many drugs [217,218], but not for all antibiotic classes. The limits of this technique are low specificity, low precision, and low sensibility. This aspect may be problematic within TDM practice, where cross-reactivity with drug metabolites and other analytes can lead to imprecision results [47]. Tsoi et al. (2019) describes a case of falsely elevated vancomycin serum concentrations as a consequence of endogenous protein interference [219].

MS coupled with high performance LC (LC-MS/MS) represents the gold standard method to quantify small molecules in biological matrices [220]. The LC system separates analytes in complex solutions using a LC column and different mobile phases. After chromatographic separation, an eluate is detected by a mass spectrometer. A specific mass/charge precursor ion is targeted for each compound and fragmented into product ions through collision with an inert gas. The advantages of this technique are many: high sensitivity, specificity, and accuracy, also for low-drug-plasma concentrations; and validation according to guidelines of the European Medicines Agency (EMA) and the Food and Drug Administration (FDA) [221,222]. The validation of LC-MS/MS methods includes the evaluation of precision, accuracy, selectivity, carryover, recovery, matrix effect, and stability. Moreover, LC-MS/MS methods are rapid, customizable, and applicable to small volumes of samples.

Several HPLC and LC-MS/MS methods have been developed for quantification of antibiotics in plasma [223,224,225,226,227,228,229,230]. HPLC-extraction methods could request more than 100 µL of plasma [225], these volumes are often not available in preterm neonates. One limit of UV methods coupled with HPLC is the possibility of interfering compounds and long sample-preparation procedures. HPLC methods for the determination of different antibiotics could use different chromatographic conditions, including isocratic mobile phases [224]. On the other hand, HPLC methods can be used in laboratories where MS technology is not available [223].

LC-MS/MS allows for high specificity and the possibility to analyze drugs and metabolites in small volumes of sample [226]. Multiplexes are bioanalytical methods validated for the determination of different antibiotics in a single chromatographic run. These methods reduce costs and analysis time, favoring rapid turnaround times [227,229]. An important advantage is that LC-MS/MS allows researchers to discriminate a drug from its metabolites [228]. This aspect is particularly important when active metabolites are analyzed. However, it is worth noting that the number of studies in which UPLC-MS/MS methods have been applied to the TDM of antibiotics in preterm neonates is still limited [231]. The reasons for this poor application probably depend on the physical and ethical concerns in obtaining blood samples from preterms. Perhaps the introduction of new micro-sampling techniques could encourage the application of TDM for antibiotics in preterm neonates.

### Microsampling Strategies Applied to PK Studies in Preterms

Sampling in preterm neonates is always a critical aspect during clinical practice. The sampling procedures for both TDM and PK studies are mainly based on venipuncture. However, this method is invasive and requires high volumes of blood. Therefore, it is characterized by poor patient compliance and is not easily and ethically applicable to preterm neonates. This is one of the main obstacles when performing TDM in clinical practice and conducting PK studies in this special population [232]. Therefore, the availability of less invasive techniques requiring smaller samples is essential in neonatal settings [47]. In this review, we focused on micro-sampling techniques applicable to preterm newborns, with particular attention being paid to the most used antibiotic therapies.

Dried Blood Spot (DBS) is a sampling method that involves taking and analyzing a dried drop of blood on paper filters. This technique has been used since the 1960s for neonatal screening by heel sampling. The use of DBS for TDM application in preterm neonates has several advantages, e.g., blood samples are collected by heel pricks and, therefore, are slightly invasive. Moreover, DBSs require small volumes of blood and are easily stored and shipped to analytical laboratories [233]. However, the methods used to measure drug concentrations on DBS require an accurate bioanalytical validation that must include several aspects such as linearity, selectivity, sensitivity, accuracy, precision, matrix effects, recovery, stability, blood-spot size, spot-punch location, blood-spot volume, and hematocrit variability [234,235]. An important limit of using DBS for TDM application is the influence of hematocrit. Blood’s viscosity affects the amount of sample collected from a spot, and the ratio of RBC/plasma can affect a drug’s concentration. Lawson and colleagues (2016) focused on the analytical bias in DBS measurements, demonstrating that the spot-punch location can significantly modify the measured concentration of an analyte [236]. Additionally, the blood-spot volume influences the drug’s concentration. In fact, a reduced drug amount has been associated with small volumes of blood [236]. How can this bias be overcome? Taking a fixed volume of blood for each spot and correcting the bias with different hematocrit levels could be a good approach. However, it is important to refer to the available guidelines for bioanalytical laboratories and TDM centers [237].

So far, several articles have been published on the use of DBSs for the TDM of antibiotics [238]. However, there are few reports involving preterm neonates. Dos Santos et al. (2022) described the first assay for the determination of gentamicin concentrations on DBS using a chromatographic method [239]. The authors compared the results of DBS determinations with plasma concentrations in a small cohort of nine neonatal patients. Acceptable results were obtained over a wide range of hematocrit values. Finally, the authors conclude that further studies are needed to validate the use of this methodology in clinical practice [239]. In 2018, Le and colleagues performed a study to test the utility and accuracy of DBS vs. plasma for evaluating the PK behavior of ampicillin in neonates [240]. In particular, 18 patients were included in this popPK analysis; 29 DBS and plasma samples were collected from each subject. A strong correlation was observed between DBSs and plasma, but the bias exceeded >15%. Therefore, the authors conclude that DBSs could be used to estimate ampicillin plasma concentration in neonates, by standardizing spot coverage on the card, drying, and storage [240].

Volumetric absorptive micro-sampling (VAMS) is a micro-sampling technique, consisting of a plastic stick with an absorbent tip made of a hydrophilic polymer. The tip absorbs a fixed volume of the sample, according to the tip size (10, 20, 30 µL). Similar to DBS, VAMS represent a valid alternative to venipuncture, since it is minimally invasive and easy to ship and to store. This technique is not yet widely used in neonatal settings, probably because the therapeutic ranges for antibiotics refer to plasma concentrations. Moreover, a complete validation of this sampling technique is required before its introduction into clinical practice [232,241,242]. An important advantage of VAMS compared to DBS is represented by the absence of the hematocrit effect [22,243]. Barco et al. (2017) developed and validated a LC-MS/MS method for the quantification of piperacillin, tazobactam, meropenem, linezolid, and ceftazidime on VAMS [244]. In particular, the authors compared this sampling strategy with DBS in terms of the impact of hematocrit on the accuracy, recovery, and matrix effect. Using VAMS, they obtained an acceptable bias (<15%) for all calibration levels at all hematocrit values analyzed. The results of this study present some limitations mainly due to the limited data available. However, this study shows that VAMS could be considered as an alternative sampling method in neonatal intensive care units [244]. The correlation between estimated plasma concentrations from VAMS and plasma concentrations obtained from classic venipuncture sampling is a crucial point to allow the implementation of VAMS utilization for TDM. Limited data are available, especially for antibiotic therapies in neonatal settings. Moorthy et al. (2020) described an application of VAMS for vancomycin determination in a pediatric clinical study [245]. They used VAMS to overcome the problem of the small blood volumes available to calculate the AUC, improving the safety and effectiveness of vancomycin administration [245].

Capillary Micro Sampling (CMS) is a sampling technique for the collection of small volumes of blood using exact volume glass capillaries from hand or heel pricks. CMS are used for TDM and PK studies in neonates in order to overcome the problems connected with traditional sampling, such as the volume of sample and infection risks [16,223]. In a recent interesting study, Parker et al. demonstrated that, in both neonatal and pediatric patients, the concentrations of meropenem from CMS and from venipuncture were comparable [246].

Micro-dialysis is a sampling method used in intensive care neonatal units for metabolic monitoring (i.e., hypoglycemia). This technique is based on the diffusion through a permeable membrane of small molecules [247]. The application of micro-dialysis for the TDM of antibiotics in neonatal settings is still not very widespread. However, an interesting advantage of this technique is that only the unbound drug fractions can pass the membrane, and this aspect is useful to exclusively measure the amount of the active drug [248].

Saliva is an alternative matrix for TDM application. It is not invasive and can be easily collected. This aspect is important for neonatal population, however, a bioanalytical validation should be performed by comparing drug concentrations in both saliva and plasma. Unfortunately, so far, few studies have been performed on antibiotic measurements in preterm newborns [249,250].

Finally, it is worth noting that, despite the number of advantages offered by these micro-sampling techniques, there are still concerns, such as analyte stability, repeated analysis, bridging correlation between plasma, and blood concentration, which must be addressed [251].

## 8. Conclusions and Future Prospective

Preterm newborns represent a special population characterized by physiological and pharmacological hallmarks that differ from full-term neonates. Therefore, tailored dosing strategies are needed when administering medications to these subjects. Unfortunately, the availability of PK studies involving neonates is affected by both ethical and physiological concerns that significantly limit sampling strategies and protocols. In the absence of clear guidelines and ad hoc studies, the management of preterms is mainly based on clinical practices, where PK parameters and dosing regimens are often extrapolated from clinical studies conducted on adults. In this context, the recent developments of pharmacometric approaches, including both popPK and PBPK, have allowed researchers to personalize the use of different drugs in preterms.

Antibiotics are largely used among preterm newborns. Therefore, they have been the object of many PK models aimed at suggesting specific dosing regimens in this population.

Therapeutic drug monitoring is a useful tool for the optimization of pharmacological treatments. This approach becomes particularly advisable in special populations, including critically ill neonates and children. In fact, developmental changes associated with growth, pathological conditions, and the presence of invasive procedures can dramatically affect the PK behavior of administered drugs, making their circulating levels often unpredictable. Advances in micro-sampling strategies, alongside the development of more sophisticated bioanalytical methods and technologies, have allowed the quantification of many drugs with their metabolites in small volumes of blood.

So far, much relevant information on the PK of antibiotics used in preterm newborns has been obtained from TDM-based pharmacometric studies. However, these suggested dosing regimens should be further validated in external prospective studies. This aspect represents a challenge to be faced in the future. Similarly, considering antimicrobial stewardship, the use of antibiotics in preterms, as well as the introduction of new molecules into clinical practice, should be carefully evaluated in terms of safety and efficacy by performing TDM and model-based predictive studies.

## Data Availability

Data sharing is not applicable to this article as no datasets were generated or analysed during the current study.

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
