# Peer review of "Use of Antibiotics in Preterm Newborns"

_antibiotics, 2022, doi:10.3390/antibiotics11091142_

Round 1
Reviewer 1 Report
To the Authors
1. I suggest to add a Table summarizing the key physiological aspects of preterm newborns and how they can affect the different pharmacokinetic phases (ADME) of antibiotics.
2. The section entitled “Physiologically based pharmacokinetic (PBPK) modelling to predict PK parameters in preterm infants” should, in my mind, numbered as 5.1 instead of 6.
3. Table 1 should be revised by adding a column summarizing the key findings of each of the quoted study.
4. Table 2 is of very limited value for readers in the present form (why did the Authors decide to focus only on LC-MS/MS bioanalythical methods for TDM of antibiotics in preterm newborns?). I suggest to replace it with a new table summarizing the key findings of studies that have dealt with the TDM of antibiotics in the preterm newborns.
5. The Authors should always use the same definition (preterm infants versus preterm newborn).
Author Response
Comments and Suggestions for Authors
Reviewer 1
- I suggest to add a Table summarizing the key physiological aspects of preterm newborns and how they can affect the different pharmacokinetic phases (ADME) of antibiotics.
Many thanks for this precious suggestion. We have now included a table (Table 1 in the revised manuscript) summarizing main physiological aspects that characterize preterm neonates alongside to their effects on pharmacokinetic processes.
- The section entitled “Physiologically based pharmacokinetic (PBPK) modelling to predict PK parameters in preterm infants” should, in my mind, numbered as 5.1 instead of 6.
Section has been now numbered as 5.1 instead of 6.
- Table 1 should be revised by adding a column summarizing the key findings of each of the quoted study.
Thanks for this suggestion. Table 1 has been modified and numbered as 2. We have included an additional column reporting the key finding of each study in terms of dosing strategy suggested.
- Table 2 is of very limited value for readers in the present form (why did the Authors decide to focus only on LC-MS/MS bioanalythical methods for TDM of antibiotics in preterm newborns?). I suggest to replace it with a new table summarizing the key findings of studies that have dealt with the TDM of antibiotics in the preterm newborns.
According to your comment, we have now removed Table 2 since it was referred to the application of LC-MS/MS bioanalythical methods for TDM of antibiotics in both adult and pediatric patients but not preterm neonates.
- The Authors should always use the same definition (preterm infants versus preterm newborn).
We do apologies for this misleading point. We have alternatively used the definitions preterm newborns and preterm neonates to avoid repetitions. Moreover, both terms gave on Pubmed the same number of results.
Reviewer 2 Report
The authors Raffele Simeoli et al., have done a great job by determining the pharmacokinetics of antibiotics in preterm newborns. There are some flaws that shall be addressed .some of them are:
1.The title of the manuscript must be changed as Use of antibiotics in Preterm.
2. In abstract line 23 the word OFF LABEL must be replaced by some suitable word.
3. Add a table showing the preterm differentiation from neonates etc. w.r.t age limits
4. Please tabulate the various antibiotics classes used in preterm must be tabulated for author’s easy understanding. Also the doses shall also be added.
5. Add some lines for the Therapeutic drug monitoring at the end.
6. Add limitations of micro sampling in preterm as venipuncture is very difficult in preterm.
7. Yet I am not a native speaker of English language but still I recommend that the English language needs touching up in a major way. The article needs to be rewritten in readable English. Many sentences are confusing, do not lead to scientific meaning, and can be found starting in lower case, and upper case can be detected in the middle of sentences without proper nouns.
Author Response
Reviewer 2
The authors Raffaele Simeoli et al., have done a great job by determining the pharmacokinetics of antibiotics in preterm newborns. There are some flaws that shall be addressed .some of them are:
1.The title of the manuscript must be changed as Use of antibiotics in Preterm.
Title of our manuscript has been changed according to your suggestion.
- In abstract line 23 the word OFF LABEL must be replaced by some suitable word.
The term “off-label” refers to use of a drug that is not included in the package insert (approved labeling) for that medication. Specifically, drugs are used off-label when administered for an unapproved indication or in an unapproved age group, dosage, or route of administration (Randall S. Stafford (2008). "Regulating Off-Label Drug Use — Rethinking the Role of the FDA". N Engl J Med. 358 (14): 1427–1429. doi:10.1056/NEJMp0802107). Therefore, the term “off-label” is specifically used to indicate drugs administered to neonates (pre- or full-terms) for whom there is not an approved indication for age group.
- Add a table showing the preterm differentiation from neonates etc. w.r.t age limits
Many thanks for this precious suggestion. We have now included a table (Table 1 in the revised manuscript) summarizing main physiological aspects that differentiate preterm from full-term neonates alongside to their effects on pharmacokinetic processes.
- Please tabulate the various antibiotics classes used in preterm must be tabulated for author’s easy understanding. Also the doses shall also be added.
Thanks for this suggestion. Table 1 has been modified and numbered as 2. We have included an additional column reporting the key finding of each study in terms of dosing regimen suggested.
- Add some lines for the Therapeutic drug monitoring at the end.
We have included some lines on TDM at the end of paragraph numbered as 7 in the revised manuscript.
- Add limitations of micro sampling in preterm as venipuncture is very difficult in preterm.
Thanks for this suggestion. We have now included at the end of paragraph 7.1 in the revised version of manuscript few sentences on the limitations of microsampling procedures.
- Yet I am not a native speaker of English language but still I recommend that the English language needs touching up in a major way. The article needs to be rewritten in readable English. Many sentences are confusing, do not lead to scientific meaning, and can be found starting in lower case, and upper case can be detected in the middle of sentences without proper nouns.
The text has been fully revised and language has been improved.
Reviewer 3 Report
In this Review article Raffaele Simeoli et al. provide the important findings in pharmacokinetics of antibiotics in preterm newborns. Describing and comparing PK/PD of different classes of antibiotics in neonates and adults.
The Review articles was well written, and the conclusions are appropriate. I believe that the information provided and described in the review are appropriate to publish in Antibiotics Journal.
Below are suggestions for minor revision of the manuscript:
1. Authors would have used appropriate words instead of using the word “worst” in introduction part first paragraph and also in the second paragraph word “Unfortunately”
2. Pharmacokinetics of antibiotics section in section 7.3 Glycopetides It wpould be nice to include the reference article for the sentence” glycopeptides inhibit bacterial cell membrane permeability”
However, I have no major concerns with this review article in its current form as well.
Author Response
Reviewer 3
In this Review article Raffaele Simeoli et al. provide the important findings in pharmacokinetics of antibiotics in preterm newborns. Describing and comparing PK/PD of different classes of antibiotics in neonates and adults.
The Review articles was well written, and the conclusions are appropriate. I believe that the information provided and described in the review are appropriate to publish in Antibiotics Journal.
Below are suggestions for minor revision of the manuscript:
- Authors would have used appropriate words instead of using the word “worst” in introduction part first paragraph and also in the second paragraph word “Unfortunately”.
Many thanks for highlighting these points. Both words have been now appropriately replaced.
- Pharmacokinetics of antibiotics section in section 7.3 Glycopetides It wpould be nice to include the reference article for the sentence” glycopeptides inhibit bacterial cell membrane permeability”
Two references have been added to support the above sentence.
However, I have no major concerns with this review article in its current form as well.